# ATLAS: Alibaba Dataset and Benchmark for Learning-Augmented Scheduling

**Zhiyun Jiang**   **Tianming Zhao**   **Chunqiu Xia**   **Albert Y. Zomaya**
University of Sydney
{zjia5638, tianming.zhao, chunqiu.xia, albert.zomaya}@sydney.edu.au

## Abstract

Learning-augmented scheduling uses ML predictions to improve decision-making under uncertainty. Many algorithms in this class have been proposed with better theoretical guarantees than the classic methods. Translating these theoretical results into practice, however, requires an understanding of real workloads. Such an understanding is hard to develop because existing production traces either lack the ground-truth processing times or are not publicly available, while synthetic benchmarks fail to represent real-world complexity. We fill this gap by introducing *Alibaba Trace for Learning-Augmented Scheduling (ATLAS)*, a research-ready dataset derived from Alibaba's Platform of Artificial Intelligence (PAI) cluster trace—a production system that processes hundreds of thousands of ML jobs per day. The ATLAS dataset has been cleaned and features engineered to represent the inputs and constraints of non-clairvoyant scheduling, including user tags, resource requests (CPU/GPU/memory), and job structures with ground-truth processing times. We develop a prediction benchmark reporting prediction error metrics, along with feature importance analysis, and introduce a novel multi-stage ML model. We also provide a scheduling benchmark for minimizing the total completion time, max-stretch, and makespan. ATLAS is a reproducible foundation for researchers to study learning-augmented scheduling on real workloads, available at `https://github.com/zhiyunjiang0810/non-clairvoyant-with-predictions`.

## 1 Introduction

Modern computing systems have to schedule millions of jobs across without knowing job sizes (i.e., processing time) at submission, a challenge known as non-clairvoyant scheduling. As job sizes are unknown at arrival, the scheduler cannot implement an optimal clairvoyant strategy such as SRPT for total completion time; consequently, non-clairvoyant algorithms achieve suboptimal scheduling performance (Motwani et al., 1994). Learning-augmented algorithms address this performance degradation by incorporating ML job size predictions into online algorithms, improving performance while maintaining worst-case guarantees (Kumar et al., 2018). This framework applies to many domains, and now has grown into an active community (Lindermayr & Megow, 2022).

To illustrate the effect of predictions, consider the single-machine scheduling to minimize total completion time $\sum C_j$, where $C_j$ is the completion time of job $J_j$, in Figure 1. Suppose we have four jobs released at $r_j$ with unknown sizes $p_j^*$. Without job predictions, First-In-First-Out (FIFO) runs jobs in arrival order, yielding a total completion time of 51. Round Robin(RR), another good default for non-clairvoyant scheduling (Motwani et al., 1994), achieves $\sum C_j$ of 46.3. With predictions $\hat{p}_j$, Shortest Predicted Job First (SPJF) runs jobs by job size predictions. Even imperfect predictions add value when they roughly reflect relative job order. With true sizes, Shortest Remaining Processing Time (SRPT) is optimal for this problem (Schrage, 1968), yielding a total completion time of 33. Beyond $\sum C_j$, other interesting objectives include maximum stretch, which measures fairness via the maximum ratio of job response time to size $\max_j \frac{C_j - r_j}{p_j^*}$, and makespan $C_{\max} = \max_j C_j$, representing the completion time of the last job, a classic objective in parallel-machine scheduling (Zheng et al., 2023). For each specific objective, recent theoretical work has developed learning-augmented algorithms with provable guarantees (Zhao et al., 2024; Lattanzi et al., 2020; Kumar et al., 2018).

Figure 1: A toy example showing the comparison of online/offline scheduling algorithms on a single machine with four jobs arriving at $0, 0, 1, 2$ with job sizes $4, 10, 1, 3$ and job size predictions $3, 11, 2, 1$. Despite prediction errors, SPJF, a widely known learning-augmented algorithm, achieves a total completion time of 37, remaining close to SRPT's optimum of 33 and outperforming both Round Robin's 46.3 and FIFO's 51.

However, real production clusters often violate core assumptions underlying these theoretical models. In practice, jobs execute as multi-step workflows (e.g., preprocessing before training) where early-stage failures can terminate the sequence. Furthermore, hardware is heterogeneous, and arrivals are stochastic, undermining standard analyses (Weng et al., 2022). These disconnects between theory and practice raise a central question: *How well do learning-augmented schedulers perform in real-world environments?* We address the problem with **Alibaba Trace for Learning-Augmented Scheduling (ATLAS)**, the first dataset for learning-augmented scheduling derived from PAI production clusters, covering over 730,000 jobs with complete execution histories and resource profiles.

**Issues with current datasets and benchmarks.** First, existing production traces offer limited data for training and evaluating predictors for job processing times. Google's Borg traces (Tirmazi et al., 2020) normalize processing times and obfuscate job identities, removing rich context like user patterns, job types, resource requests, and historical behavior. Azure public datasets (Cortez et al., 2017) and Microsoft's Virtual Machine (VM) allocation traces (Lu et al., 2017) focus primarily on VM provisioning, exposing utilization rates while omitting job structures or exact completion times. The Alibaba trace (Weng et al., 2022) provides job structures but was designed for workload characterization rather than scheduling evaluations. Second, most theoretical studies rely exclusively on synthetic workloads (Zhao et al., 2022; Benomar & Perchet, 2024), limiting job sizes to standard exponential, Pareto, or uniform distributions that miss the complex patterns found in real systems. Third, the field lacks a standardized evaluation benchmark: a clear, reproducible specification of (a) the scheduling framework (online/offline, (non-)preemptive, number of machines), (b) how predictors are trained and validated, and (c) how results are reported and normalized. Consequently, different studies adopt incompatible problem formulations, metrics, and experimental setups, such as work by Fan et al. (2022); Im et al. (2023); Bampis et al. (2023), making cross-paper algorithm comparisons difficult. Furthermore, many overlook temporal constraints (training on past, testing on future), failing to restrict features to historical information, or skip calibration–test separation, risking information leakage that violates non-clairvoyant assumptions (Kapoor & Narayanan, 2023).

## 1.1 OUR WORK

**The ATLAS Dataset.** ATLAS transforms raw production traces from Alibaba's Platform of Artificial Intelligence cluster into a dataset designed for scheduling research. The dataset contains completed ML jobs collected from a cluster with over 6,500 GPUs across 1,800 machines. The overview and statistics are shown in Table 1. ATLAS dataset has three defining characteristics:

*(1) Non-clairvoyant dataset:* Our dataset is non-clairvoyant, where models access only information available at submission time, such as resource plans (resource requests) for CPU, memory, GPU, and instance counts for each task, and identity fields needed to build signatures such as user, group, and workload. We exclude post-execution metrics, actual resource usage, utilization rates, and machine placements, ensuring the ATLAS dataset is research-ready, which replicates a real scheduling environment. *(2) Complete ground-truth job sizes:* Unlike other production traces that omit or normalize runtime information for privacy reasons, ATLAS provides verified processing times for all tasks and jobs. Jobs, with a mean of 1.5 hours, range from 3 seconds to 7 days, with 73.8% completing within one hour and 5.7% exceeding six hours, showing a full spectrum of production workloads. *(3) Rich workload diversity:* ATLAS dataset encompasses jobs with resource requirements spanning from single-CPU tasks to distributed jobs using multiple GPU and GPU clusters, reflecting realistic heterogeneity in CPU/GPU/memory requests. The dataset includes 74.4% single-instance jobs, multi-instance jobs, and distributed jobs with up to 1,050 instances, capturing job scale complexity.

Table 1: ATLAS dataset contains jobs from a two-month Alibaba trace with selected submit-time known and engineered without data-leakage features. ✓ = prediction feature; ● = ground-truth label; × = excluded.

| Source | Field | Use | Description | Dataset Statistics | |
|---|---|---|---|---|---|
| **Job** | Submission time | ✓ | Release time $r_j$; enables temporal patterns | **Dataset splits** | |
| | User ID | ✓ | Anonymized submitter (1,314 users) | Training | 512,649 (70%) |
| | Processing time | ● | Label $p_j^* = \min_t s_t - \max_t e_t$ | Validation | 109,853 (15%) |
| | | | | Test | 109,853 (15%) |
| **Task** | Task count | ✓ | Number of roles $[1, 20]$ (median: 1) | **Job size** | |
| | Planned CPU | ✓ | $\sum_t n_t r_{t,1}$ $[0, 810K]$ (median: 6) | Mean | 5,382 s (1.5 h) |
| | Planned GPU | ✓ | $\sum_t n_t r_{t,2}$ $[0, 40K]$ (median: 1) | Median | 663 s (11 min) |
| | Planned Memory | ✓ | $\sum_t n_t r_{t,3}$ $[0.4, 47K]$ GiB (median: 29) | Std. Dev. | 17,095 s |
| | Instance count | ✓ | Total parallelism $\sum_t n_t$ $[1, 1050]$ | Range | [3 s, 626,384 s] |
| **Group-tag** | Group tag | ✓ | Semantic cluster (65% recurring tasks) | **Job duration** | |
| | Workload tag | ✓ | Application type when known | <10 min | 355,328 (48.5%) |
| | GPU spec | ✓ | Submit-time constraint (V100/P100/T4) | 10 min–6 h | 335,121 (45.8%) |
| | Recurrence count | ✓ | Historical submission frequency | >6 h | 41,906 (5.7%) |
| **Engineered** | CPU per GPU | ✓ | Resource ratio $r_{cpu}/r_{gpu}$ | **Job-scale** | |
| | Memory per GPU | ✓ | Resource ratio $r_{mem}/r_{gpu}$ | Single-instance(1) | 544,881 (74.4%) |
| | Distributed flag | ✓ | Binary indicator ($n > 1$) | Small (2–10) | 93,951 (12.8%) |
| | User history | ✓ | Mean/std of user's past submissions | Medium (11–100) | 86,380 (11.8%) |
| | Group history | ✓ | Mean/std/count of group's past runs | Large (>100) | 7,143 (1.0%) |
| | Scale category | ✓ | Composite job-scale classification | | |
| **Excluded** | Assigned GPU | × | Actual placement (V100/P100/T4/Misc) | **GPU categories** | |
| | Instance IDs | × | Worker/container identifiers | No GPU | 17,452 (2.4%) |
| | Sensor metrics | × | Runtime CPU/GPU/memory utilization | Single GPU | 554,179 (75.7%) |
| | Network usage | × | Bandwidth and I/O measurements | Multi-GPU | 110,474 (15.1%) |
| | Machine specs | × | Hardware configuration details | GPU cluster | 50,250 (6.9%) |

**The LASched Benchmark.** Built on top of ATLAS, LASched (Learning-Augmented Scheduling Benchmark) provides a standardized evaluation for the *Prediction Task* on job sizes and *Scheduling Task* of jobs with job size predictions. The prediction benchmark implements multiple baseline models and evaluation metrics, showing that jobs exhibit recurring patterns that can be leveraged for accuracy. LASched enforces leakage-safe construction and restricts features to past-only information available at decision time, preventing future leakage. The scheduling benchmark evaluates classic and learning-augmented schedulers on three metrics—total completion time, maximum stretch, and makespan. With a scheduling benchmark including various baselines, using job size predictions to schedule, the objective values are normalized against the optimum. Together, ATLAS and LASched form a complete platform that enables researchers to develop, evaluate, and compare learning-augmented scheduling methods on real production workloads with reproducible results.

## 2 THE **ATLAS** DATASET

### 2.1 DATA SOURCE AND FORMALIZATION

**Alibaba PAI–2020 Trace.** **ATLAS** is built on the publicly available Alibaba PAI–2020 GPU-cluster trace, which captures two months of MLaaS activity on a large heterogeneous GPU cluster over 6,500 GPUs across ∼1,800 machines (Weng et al., 2022). The trace captures the job life-cycle, including submission, queuing, scheduling, and execution. The trace is relational and organized hierarchically into jobs, tasks, and instances, consistent with established system frameworks such as Google Borg (Verma et al., 2015) and Facebook's Hadoop workloads (Zaharia et al., 2008). Users submit ML jobs through frameworks (e.g., TensorFlow, PyTorch, Graph-Learn); each job is assigned to a scheduler (Fuxi), which translates it into multiple tasks with different roles, e.g., worker, parameter-server (PS), and then instantiates them into Docker containers that are distributed across multiple machines based on resource availability and locality requirements (Weng et al., 2022). Once started, jobs run to completion without preemption. The trace contains only start and end timestamps for each instance, not suspension or resumption events. PAI's monitoring collects per-instance system metrics: CPU/GPU utilization and host/GPU memory, at every 15 seconds via daemon agents that query the Linux kernel and NVIDIA's NVML; the release data also includes machine-level statistics such as network receive throughput. Figure 2 summarizes the job schema, showing hierarchical structure, and the relationship between jobs, planned resources, observed utilization, and machine specifications. We formalize the PAI trace columns so that time semantics,

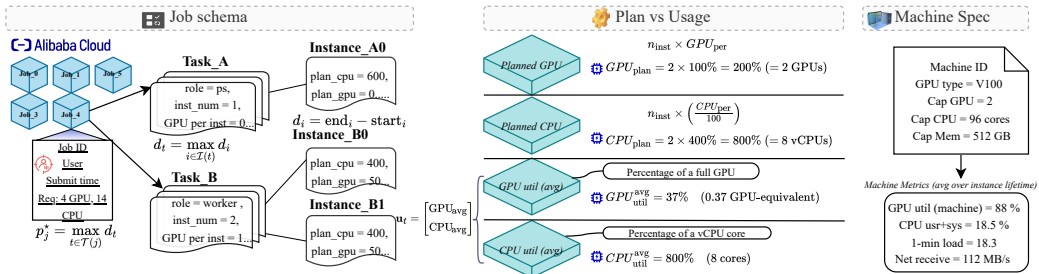

Figure 2: Structure of the Alibaba PAI-2020 GPU cluster trace. Jobs decompose hierarchically into tasks and instances with per-instance resource requirements for GPU and CPU. Job processing time $p_j^* = \max_{t \in \mathcal{T}(j)} d_t$ equals the maximum task duration. The center panel shows planned versus actual resource usage, revealing users requesting more resources than they actually use. Instances can request fractional resources (e.g., 0.5 GPU), supporting resource sharing across multiple jobs. Machine specifications and utilization metrics appear on the right. Task B illustrates gang scheduling $g_t = 1$, where more than one instances start at the same time.

label construction, and reproducibility checks are unambiguous, whereas prior analyses of this trace reported workload behavior without a unified mathematical specification.

**Job life-cycle: arrivals, queuing, launching.** We formalise job life-cycle mathematically as follows. A job $J_j$ arrives at time $r_j$ and consists of tasks $t \in \mathcal{T}(j)$. Each task $t$ declares a per-instance demand vector $\mathbf{r}_t = (r_{t,1}, r_{t,2}, r_{t,3})^\top \in \mathbb{R}_+^3$ representing per-instance GPU, CPU, and memory requests. To be specific, a distributed training worker might request $\mathbf{r}_t = (1, 8, 32)$ for 1 GPU, 8 CPUs, and 32GB memory. Also, each task includes an instance count $n_t$, and constraints: an admissible GPU-type set $\Gamma_t$, a gang flag $g_t \in \{0, 1\}$, and an optional locality flag $\ell_t \in \{0, 1\}$. The cluster comprises machines and each machine $m$ with capacity vector $\mathbf{c}_m = (c_{m,1}, c_{m,2}, c_{m,3})^\top \in \mathbb{R}_+^3$ corresponding to GPU, CPU and memory capacity. While the job waits, the scheduler seeks, for each task $t$, the earliest time $\tau \geq r_j$ at which its $n_t$ instances admit a feasible placement. Let $\mathcal{I}(t)$ be the instances of $t$; let $t(i)$ denote the task of instance $i$; let $x_{i,m}(\tau) \in \{0, 1\}$ indicate that instance $i$ is assigned to machine $m$ at time $\tau$ (i.e., $x_{i,m}(\tau) = 1$ if instance $i$ runs on $m$ at time $\tau$, 0 otherwise). Feasibility requires, for every machine $m$, the resource capacity constraint: $\sum_i x_{i,m}(\tau) \mathbf{r}_{t(i)} \leq \mathbf{c}_m$ Where the inequality holds for each resource dimension: GPU, CPU and memory. If $g_t = 1$ then all instances of $t$ must start together, i.e., $\sum_m \sum_{i \in \mathcal{I}(t)} x_{i,m}(\tau) = n_t$; if $\ell_t = 1$ then all instances of $t$ must be co-located on some machine $m_t$, i.e., $\sum_{i \in \mathcal{I}(t)} x_{i,m_t}(\tau) = n_t$; for GPU-type admissibility, let $g(m)$ denote the GPU type of machine $m$. We enforce $x_{i,m}(\tau) = 0$, whenever $g(m) \notin \Gamma_t, \forall i \in \mathcal{I}(t), \forall m, \forall \tau$, i.e., instances are ineligible for machines of the wrong GPU type. Define the task-ready time $s_t := \inf\{\tau \geq r_j : \text{ a feasible placement for } t \text{ exists at } \tau\}$. The job's start time is the earliest task launch $s_j = \min_{t \in \mathcal{T}(j)} s_t$, and the queuing delay is $q_j = s_j - r_j$. In the PAI trace, the job table's start time stores the submission time $r_j$, while task and instance tables record the realized launches and finishes. In production, PAI uses reserving-and-packing scheduling: it reserves high-end V100/V100M32 (NVLink) nodes for high-GPU or strict gang/locality tasks, and packs lower-GPU tasks onto T4/older 'Misc' machines via fractional-GPU sharing (Weng et al., 2022).

**Job processing time.** For every instance $i$ we record its start and end time $(s_i, e_i)$ and the duration as $d_i = e_i - s_i$. For each task $t$ with instances $\mathcal{I}(t)$, define $s_t = \min_{i \in \mathcal{I}(t)} s_i$, $e_t = \max_{i \in \mathcal{I}(t)} e_i$, and $d_t = e_t - s_t$. Let $S = \min_t s_t$ and $E = \max_t e_t$. The job-level processing time is:

$$p_j^* = E - S$$

For any task $t$, $d_t = e_t - s_t \leq E - S = p_j^*$, hence $\max_t d_t \leq p_j^*$. Prior work by Weng et al. (2022) trains a regressor on instance records and predicts a per-instance duration $\hat{d}_i$; scheduling and error evaluation are performed at the instance granularity. When a value is summarized for a task, it is the mean across that task's instances, $\bar{d}_t = \frac{1}{|\mathcal{I}(t)|} \sum_{i \in \mathcal{I}(t)} d_i$. In contrast, we define the job-level label without using the mean task duration: $\max_t e_t - \min_t s_t$, which matches the classic fork–join completion rule: a task completes when all of its instances complete, and a job completes when all of its tasks complete (Blumofe & Leiserson, 1999; Ko & Serfozo, 2004). Under gang scheduling

Table 2: Three representative job examples from the ATLAS submit-time dataset, and the processing time serves as the prediction label representing the uninterrupted execution duration.

| Job Type | pai_job_table | | | | pai_task_table | | | | pai_group_tag_table | | | Processing Time |
|---|---|---|---|---|---|---|---|---|---|---|---|---|
| | job_name | user_id | submit_time | tasks | instances | cpu (%) | gpu (%) | mem (GB) | group_tag | workload | recurrence | (Label) |
| Small Inference | d7eb43b8... | 5b1345f0... | 09:23:15 | 1 | 1 | 600 | 25 | 29.3 | 6c0d75d7... | - | 47 | 136 s |
| Distributed Training | 84afa920... | d4d51aca... | 10:45:30 | 2 | 26 | 15,100 | 625 | 52.0 | aba828a1... | ctr | 12 | 9,493 s |
| Large Scale | e6145fb3... | df2899e2... | 14:12:45 | 2 | 105 | 55,000 | 4,000 | 2,050.8 | e9d4c564... | - | 3 | 44,632 s |

Figure 3: SRPT scheduling of three real PAI jobs. The table shows job characteristics. The timeline illustrates how jobs are allocated: Job 1 (small inference) completes quickly on Machine 1; Job 2 (distributed training) runs tasks in parallel on Machines 2 and 3; Job 3 (large scale) has Task 1 starting immediately on Machine 4 while Task 2 waits for Machine 3, demonstrating the impact of resource heterogeneity on scheduling decisions.

$(g_t = 1)$, $\sum_m \sum_{i \in \mathcal{I}(t)} x_{i,m}(\tau) = n_t$, instances of a task start together $(s_i = s_t)$, so $d_t = \max_i d_i$ and per-instance predictions can be used as task size directly. Weng et al. (2022) report that $85\%$ of task instances in the PAI trace require gang scheduling; in the remaining cases, instances may start at different times, such as one at 0s and one at 50s, but both lasting 100s. The same policies apply to job level. We propose $p_j^\star$ that is reconstructed from timestamps in the task table, not just using the mean task size, which is a more pessimistic method. Admittedly, neither definition decouples the true job demand from Fuxi's historical allocation decisions; allocation preferences and CPU-bound contention remain embedded in the recorded durations. However, the trade-off lies in safety. $\bar{d}_t \leq d_t$ filters skew systematically underestimates occupancy for gang-scheduled tasks where tail determines release. Conversely, $p_j^\star$ captures actual system constraints. We explicitly enforce robustness.

**Resource metrics and utilization.** Let resource coordinates $k \in \{1, 2, 3\}$ denote GPU, CPU, and memory, respectively. We distinguish submit-time requests from post-execution utilization metrics. Each task $t$ declares a per-instance request $\mathbf{r}_t = (r_{t,1}, r_{t,2}, r_{t,3})$ and instance count $n_t$, yielding total request $\mathbf{R}_t = n_t \mathbf{r}_t$. For job $j$ with tasks $\mathcal{T}(j)$, the submit-time request known at arrival is $\mathbf{R}_j = \sum_{t \in \mathcal{T}(j)} \mathbf{R}_t$. These submit-time quantities $\{\mathbf{r}_t, n_t, \mathbf{R}_j\}$ constitute the prediction features available at scheduling time. Post-execution, instance $i$ runs on interval $[s_i, e_i)$ with duration $d_i = e_i - s_i$ on machine $m(i)$ having capacity vector $\mathbf{c}_{m(i)}$. Let $\tilde{u}_{i,k} \in [0, 1]$ denote the time-averaged utilization fraction of resource $k$ for instance $i$; the resource–time consumed is $A_{i,k} = \tilde{u}_{i,k} c_{m(i),k} d_i$. Aggregating to tasks and jobs yields $A_{t,k} = \sum_{i \in \mathcal{I}(t)} A_{i,k}$ and $A_{j,k} = \sum_{t \in \mathcal{T}(j)} A_{t,k}$. Over reporting horizon $H > 0$, the average utilization of machine $m$ on resource $k$ is $\bar{u}_{\text{host},k}(m) = \frac{1}{H c_{m,k}} \sum_{i \in \mathcal{I}(m)} A_{i,k}$. With cluster capacity $C_k = \sum_m c_{m,k}$, the cluster-level utilization is $\bar{u}_{\text{cluster},k} = \frac{1}{H C_k} \sum_j A_{j,k}$. These realized usage metrics $\{\tilde{u}_{i,k}, A_{\cdot,k}, \bar{u}_{\cdot,k}\}$ are computed after job completion for workload characterization and data validation, not for prediction (Verma et al., 2015; Jeon et al., 2019; Weng et al., 2022).

## 2.2 DATASET DESCRIPTION

**Revised data columns.** We extract submit-time features from three of the original Alibaba's PAI joint tables to build a dataset for learning-augmented non-clairvoyant scheduling, specified by Table 1. From the *job table*, we keep job's start time and user ID as features, while task and instance timestamps are used solely to compute the ground-truth processing time $p_j^* = \max_t e_t - \min_t s_t$. The *task table* provides resource requirements (CPU, GPU, memory) and parallelism metrics (task and instance counts), aggregated to job level; we exclude assigned GPU types as these reflect post-submission scheduling decisions. The *group-tag table* contributes semantic identifiers and GPU specifications that encode submission-time constraints and recurrence patterns. The remaining tables, sensor and instance, are excluded as they contain either redundant information, post-execution metrics, or scheduling-dependent outcomes incompatible with non-clairvoyant scheduling. Table 5 illustrates data columns with real information for representative jobs derived from the ATLAS.

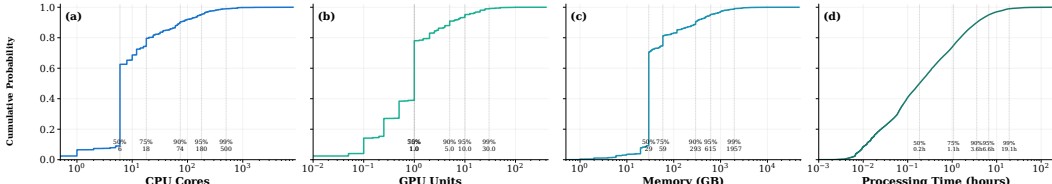

Figure 4: Resource demand and processing-time distributions in the PAI workload. Panels (a)–(c) plot cumulative distribution function (CDF) of per-job requested resources on a log-scaled x-axis with a common y-axis, Cumulative Probability. (a) total requested CPU cores, (b) total requested GPUs, and (c) total requested memory (GiB). Requests are computed as per-instance plans multiplied by instance count and aggregated per job. (d) shows the CDF of per-job $p_j^*$, defined as the elapsed time from the first task launch to the job's completion.

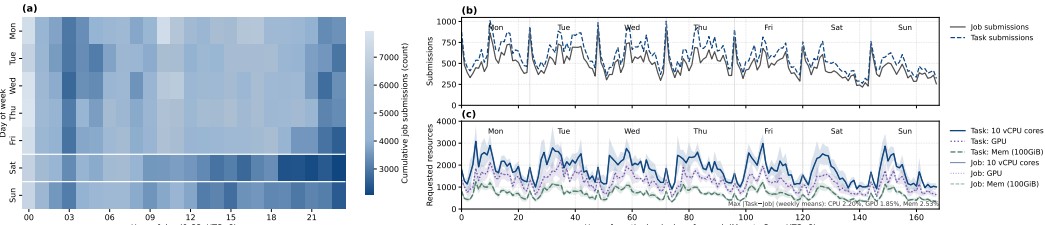

Figure 5: Temporal patterns of submissions and planned resource demand.(a) Heatmap of cumulative job submissions by weekday (rows) and hour (columns). (b) Weekly mean job submissions per hour over complete Monday–Sunday weeks (jobs and tasks).(c) Weekly means of total planned requests from obtained from per-task, per-instance plans $n_t \mathbf{r}_t$ (CPU, GPU, memory); shaded bands show 95% confidence intervals across weeks.

**Case study.** Figure 3 illustrates heterogeneous-GPU scheduling on representative PAI jobs. For this case study we assume SRPT with known processing times; the goal is to illustrate how trace semantics and resource constraints impact on execution. The small inference job (136 s) is prioritized and finishes quickly, minimizing its impact on throughput. The distributed training job, though longer, exploits parallelism with tasks running concurrently on different nodes. The large-scale job, arriving at t=100, shows how task-level scheduling adds flexibility: Task 2 starts immediately on available machine 4, while Task 1 queues for a V100 due to GPU-type locality. This yields asymmetric task completions (44,732 s vs. 54,125 s), with the job's finish dictated by the slower task.

## 2.3 Work load characterization

**Data cleaning.** We keep only *terminated* jobs/tasks and drop rows with missing timestamps or non-positive instance counts. In the trace, timestamps are converted to UTC+8, and for any time–series statistics we restrict to complete Monday–Sunday weeks, discarding partial weeks to avoid edge cases; this dataset choice improves comparability and reduces noise. We reproduced the instance–anchored plot in Weng et al. (2022) for verification, but our scope is on job– and task–level.

**Heavy-skewed distribution.** The PAI workload exhibits extreme heterogeneity at the job level, with distributions showing severe right-skew. Skewness is 11.02 for processing time and coefficient of variation is larger than 3 for resources. As shown in Figure 4, resource requests size 3–5 orders of magnitude: CPUs (0–8,100 cores), GPUs (0–400 units), and memory (0.4 GB–47 TB). Processing times vary from seconds to days, with the 99th percentile (19.1 hours) being 106× the median: 11 minutes. The workload stratifies into distinct scales: 74% single-instance jobs versus distributed jobs with up to 1,050 instances, and 66% tiny jobs (median 7-minute runtime) versus 1% massive jobs (median 77-minute runtime). This compound heterogeneity—where a small fraction of jobs dominate resource consumption—necessitates log transformation, which reduces skewness from 11.02 to 0.17 and scale-aware modeling for effective learning-augmented scheduling.

**Temporal pattern.** The trace exhibits strong diurnal/weekly regularity: weekends have fewer submissions, and late–night hours show higher arrivals, consistent with prior production studies (Tirmazi et al., 2020; Reiss et al., 2011). Whereas Weng et al. (2022) visualize a single week, we report the weekly average across complete weeks, shown in Figure 5, which smooths episodic spikes and

yields a more stable signature. Concretely, letting $h \in \{0, \ldots, 167\}$ index hour–of–week and $w$ index weeks, we average hourly request series (e.g., task–anchored $T_w(h)$ and job–anchored $J_w(h)$ formed from planned totals $\mathbf{R}_t = n_t \mathbf{r}_t$) as $\bar{T}(h) = |\mathcal{W}|^{-1} \sum_{w \in \mathcal{W}} T_w(h)$ and analogously for $\bar{J}(h)$. In our data, $\bar{T}(h)$ and $\bar{J}(h)$ are nearly identical, reflecting that most tasks launch within the job's start hour, while the averaged curves remain less sharp, but more robust than a single-week analysis.

## 3 BENCHMARK

### 3.1 PREPARATION

**Data pre-processing.** We use the ATLAS dataset, which constructs a submit-time, job-level table by joining the job, task, and group-tag relations from ALIBABA PAI trace and retaining only terminated records. Timestamps are parsed as seconds, and empty rows are removed. For each task $t$ with instances, we set $s_t = \min_i s_{t,i}$, $e_t = \max_i e_{t,i}$, and define the job processing time as $p_j^* = \max_t e_t - \min_t s_t$. Jobs with $p_j^* \leq 0$ are discarded. The submission time $r_j$ anchors chronology and all causal features. Submit-time resource declarations are aggregated per job by summing the times of per-task plans multiplied by their multiplicity, and details are in Table 1. We join the group tag, user identifier, workload tag, and requested GPU specification via the instance identifier; assigned hardware and any post-submission outcomes are excluded. To avoid leakage, we split by $r_j$: the earliest $70\%$ for training, next $15\%$ for validation, and final $15\%$ for testing. Before training, we run simple checks and use log-transformed $p_j^\star$ as the prediction target to stabilize heavy tails.

**Feature engineering.** From raw PAI ATLAS data. including 13 columns, we engineer 40 additional features into 53 data frame columns, filtering to 33 model features after removing identifiers and intermediates. All encoders and statistics use training data only. (1) *Resources:* log-transformed totals (CPU, GPU, memory, instances, tasks) and per-instance ratios (CPU/instance, GPU/instance, memory/instance, tasks/instance), addressing the heavy-tailed PAI distributions (Weng et al., 2022) (Figure 4). (2) *Temporal:* sine–cosine hour-of-day encoding to preserve cyclic continuity (Jiang & Zhang, 2009), plus day-of-week and weekend flags. (3) *Recurrence signatures:* concatenate user, group, workload, and decile-bucketed resources; match to historical executions and attach the same train-only statistics such as mean, median, quartiles, standard deviation, and counts. (4) *Historical:* strictly causal, submit-time–ordered expanding statistics for users and groups on $y = \log(1 + p^\star)$—cumulative means, counts, and exponentially weighted moving averages (span=10)—all with one-step lags via `shift(1)`; low-support groups use Empirical Bayes shrinkage ($\lambda = 5$) toward the training-set mean. (5) *Categorical:* user, group, workload, and GPU specification are label-encoded from the training set vocabulary with unseen values mapped to $-1$.

**Ablation study and overfitting analysis.** The ablation study, using LightGBM, reveals that workload recurrence and group-level execution patterns are the dominant predictive signals (+20.2% over a resource-only baseline), while individual user behaviors provide secondary refinement. The results validate our benchmark design and demonstrate that all using features contribute meaningfully to prediction accuracy, with group-level patterns generalizable across users and resource features transferable across datasets. Our overfitting analysis shows a minimal 1.1% Cov@25% gap between 5-fold cross-validation training and test results, indicating negligible overfitting to the training data. While performance naturally drops for unseen users, a 5.8% gap, due to missing user-specific history, the model maintains robust accuracy by relying on generalizable group and resource features.

### 3.2 PREDICTION TASK

**Prediction models.** We model $y_j = \log(1 + p_j^\star)$, where $p_j^\star$ is from earliest task start to latest task end, from submit-time features $\mathbf{x}_j$ using gradient boosting with validation-based calibration. We include four baselines (Single-Stage, CRR++, Scale-Experts, Recency-Fixed) that use rolling history without task-signature features, and six advanced methods specified in Appendix D: (1) *Conformal quantile regression* (CQR) training quantile regressors at $\alpha \in \{0.1, 0.5, 0.9\}$ with Ridge-blended final predictions (Romano et al., 2019); (2) *Isotonic calibration* ensuring monotonic mapping adapted for regression calibration (Zadrozny & Elkan, 2002; Kuleshov et al., 2018); (3) *Meta-stacking* combines diverse base models (L2, regularized, quantile, Huber) via gradient boosting on validation

predictions (Wolpert, 1992); (4) *Gated experts (two-stage)*: a classifier routes examples to capacity-matched regressors and aggregates them by soft probabilities (Jordan & Jacobs, 1994); (5) *Weighted recency* uses exponential time-decay $w_t = \exp(-\lambda(T-t))$ across multiple temporal windows with MAE-weighted blending for drift adaptation Gama et al. (2014); (6) *Historical Recency-Aware with Shrinkage* (HRAS) uses per-signature means with EB shrinkage to stabilize predictions for rare user-group-resource patterns. All calibrators fit exclusively on validation data following honest prediction principles (Wager & Athey, 2018), with LightGBM (Ke et al., 2017) as primary regressor.

**Multi-stage predictor.** Baselines without task-signature features reached only 19.5–20.2% Cov@25% (Table 3), motivating our calibration-centric designs. We therefore evaluate all methods within one leakage-free framework in Algorithm 1, cooperating ML prediction methods above.

---

**Algorithm 1** Multi-Method Job Duration Prediction

---

1: **Input:** time-ordered splits by submit time $r_j$: $D_{train}$ 70%, $D_{validation}$ 15%, $D_{test}$ 15%
2: **Features:** $\mathbf{x}_j = [\mathbf{x}_r, \mathbf{x}_t, \mathbf{x}_h, \mathbf{x}_c]$
3:      $\mathbf{x}_r$: job totals from task table (CPU/GPU per-inst%$\rightarrow$ counts$\times$inst), $\log$-totals, per-inst ratios
4:      $\mathbf{x}_t$: $\sin(2\pi h/24)$, $\cos(2\pi h/24)$, day-of-week
5:      $\mathbf{x}_h$: user/group histories with within-group `shift(1)`; time-since-last-submit
6:      $\mathbf{x}_c$: categorical (user, group, workload, gpu_type_spec) encoded from $D_{tr}$ only
7: **Target:** $y_j = \log(1 + p_j^*)$, where $p_j^* = \max_t e_t - \min_t s_t$ (excludes queuing)
8: **Stage 1 (train on $D_{tr}$):**
9:      Quantiles: $Q_\alpha \leftarrow$ LGBM (quantile), $\alpha \in \{0.1, 0.5, 0.9\}$
10:      Regressors: $\{M_k\} \leftarrow$ LGBM with $\{\ell_2, \text{Huber}, \text{regularized}\}$
11:      Two-Stage: classifier $C$ on three duration classes via $\mathcal{Q}_{30,70}(y)$; expert $E_c$ per class
12:      Recency: $R_{\text{full}}$ (time-decay), $R_{50}$, $R_{20}$ (most recent 50%, 20%)
13:      Signatures: train-only stats (median/quantiles/count); EB shrinkage $\bar{\mu}_s = \frac{n_s \mu_s + \lambda \mu_0}{n_s + \lambda}$ ($\lambda$=5)
14: **Stage 2 (calibrate on $D_{va}$):**
15:      CQR: on log-target let $r = \max(Q_{0.1} - y, \ y - Q_{0.9})$, $k = \mathcal{Q}_{60}(\max(r, 0))$;
16:         set bounds $L = Q_{0.1} - k$, $U = Q_{0.9} + k$ (used as features; clip $Q_{0.5}$ to $[L, U]$)
17:      Blender: $\beta \leftarrow \text{Ridge}([L, Q_{0.5}, U, \text{clip}(Q_{0.5}), M_{\ell_2}, \text{priors}])$
18:      Isotonic: $\phi \leftarrow$ monotone fit of $(Q_{0.5}, y)$ (log domain, clipped)
19:      Meta: $\mathbf{f}_{\text{meta}} = [M_k, \text{dispersion}(M_k), \text{context}]$; $\psi \leftarrow$ LGBM on $\mathbf{f}_{\text{meta}}$
20:      Recency: $\omega_w \propto (\text{MAE}_{\text{va}}(R_w) + \epsilon)^{-1}$; normalize $\sum_w \omega_w = 1$.
21: **Stage 3 (predict on $D_{te}$):**
22:      CQR: $\hat{y} = \beta([L, Q_{0.5}, U, \text{clip}(Q_{0.5}), M_{\ell_2}, \text{priors}])$; also report $[L, U]$
23:      Isotonic: $\hat{y} = \phi(Q_{0.5})$    ;    Meta: $\hat{y} = \psi(\mathbf{f}_{\text{meta}})$
24:      Two-Stage: $\hat{y} = \sum_c \pi_c E_c(\mathbf{x})$, with $\pi_c = P(c \mid \mathbf{x})$ from $C$
25:      Recency: $\hat{y} = \sum_{w \in \{\text{full}, 50, 20\}} \omega_w R_w(\mathbf{x})$
26:      HRAS: $\hat{y} = \bar{\mu}_s$; else group EB prior; else global mean
27: **Output:** $\hat{p}_j^* = \exp(\hat{y}_j) - 1$ for all methods

---

### 3.3 SCHEDULING TASK

**Implementation Setup.** LASched evaluates objectives under following settings: for **total completion time** ($\sum_j C_j$), a single machine with online arrivals and preemptive scheduling (jobs $J_j$ released at times $r_j$); for **makespan**, $m$ parallel machines with batch release (all jobs at time 0) and non-preemptive; and for **max-stretch**, a single machine with online arrivals and preemption to capture fairness and prevent starvation of large jobs. Unlike prior work on the original dataset (Weng et al., 2022), which exploits recurring task-level patterns, we study job-level scheduling with imperfect predictions across all job types, scaling from single-machine to thousand-machine clusters.

**Baseline Algorithms.** For non-clairvoyant baselines, we use FIFO as the online default (Weng et al., 2022), RR, which shares capacity equally among active jobs (Motwani et al., 1994), and LAS (Least-Attained-Service), which prioritizes the job that has received the least service so far (Nuyens & Wierman, 2008). For clairvoyant baselines, which serve as offline performance bounds, we employ SRPT, the optimal for minimizing total completion time (Schrage, 1968), and SJF (Shortest

Job First). For the multi-machine makespan objective, we evaluate LPT (Longest Processing Time), which balances load by assigning the largest job to the least-loaded machine (Della Croce & Scatamacchia, 2020), alongside SPT (Shortest Processing Time) and a Random assignment baseline.

**Scheduling Algorithms.** These algorithms integrate predictions $\hat{p}_j$ generated by our prediction benchmark models (e.g., CQR, TwoSt) into online decision-making. For **total completion time**, we evaluate SPJF and PRR (Preferential Round-Robin). PRR is a robust mechanism that reserves a processor share $\lambda$ for the job with the smallest $\hat{p}_j$ while distributing the remaining rate $(1 - \lambda)$ equally among all jobs (Kumar et al., 2018). For **max-stretch**, we evaluate SPRPT (SRPT using predicted remaining work) and EDF-P, an Earliest-Deadline-First policy that schedules based on predicted deadlines $d_j = r_j + S_{\mathrm{adv}} \cdot \hat{p}_j$. For **makespan**, we substitute true sizes with predictions to create LPPT (Longest Predicted Processing Time) and SPPT (Shortest Predicted Processing Time), prediction variant of SPT, evaluating how prediction errors impact scheduling policies.

### 3.4 EVALUATION

**Prediction error.** From theoretical study, $\eta = \max_{1 \le j \le n} \max\{\frac{p_j^*}{p_j}, \frac{p_j}{p_j^*}\}$ and $L_1 = \sum_{j=1}^{n} |\hat{p}_j - p_j^*|$ (Kumar et al., 2018; Zhao et al., 2022) are often reported. The community moves toward building a portfolio of metrics rather than a single number (Ahmed et al., 2022), and we propose diverse empirical error metrics for the prediction task. We present Root Mean Squared Logarithmic Error $\mathrm{RMSLE} = \sqrt{\frac{1}{n} \sum_{j=1}^{n} \left(\ln(1 + \hat{p}_j) - \ln(1 + p_j^\star)\right)^2}$, which de-emphasizes large outliers in heavy-tailed job distributions (Soysal & Streit, 2021). Operational tolerance is captured by Coverage at $\tau$, the fraction of jobs predicted within a relative error $\tau$; we report $\tau \in \{0.25, 0.50\}$: $\mathrm{Cov@}\tau = \frac{100}{n} \sum_{j=1}^{n} \mathbf{1}\left(\frac{|\hat{p}_j - p_j^*|}{p_j^*} \le \tau\right)$ (Minku & Yao, 2013). Finally, to assess ranking quality, we report Spearman's rank correlation $\rho$ between $\hat{p}_j$ and $p_j^\star$ (Pearson correlation) (Bedő & Ong, 2016).

**Scheduling Performance.** We evaluate algorithms via empirical competitive ratios against optimal solutions or tight bounds. For **total completion time** $1|r_j, \mathrm{pmtn}| \sum C_j$, we normalize by SRPT: $\rho_{\mathrm{TC}} = \sum_j C_j^{\mathrm{ALG}} / \sum_j C_j^{\mathrm{SRPT}}$. For **makespan** $P||C_{\max}$, we use McNaughton's preemptive bound $\mathrm{OPT}_{\mathrm{pre}} = \max\{\sum_j p_j^*/m, \max_j p_j^*\}$ as baseline: $\rho_{\mathrm{MS}} = C_{\max}^{\mathrm{ALG}}/\mathrm{OPT}_{\mathrm{pre}}$. While non-preemptive makespan is NP-hard, LPT empirically achieves near-optimal performance ($\rho_{\mathrm{MS}} \approx 1$) on our instances. For **max-stretch** $1|r_j, \mathrm{pmtn}| \max_j S_j$, we obtain $S^*$ by bisection on $S$ with EDF-feasibility (Harchol-Balter, 2013), then run EDF at $S^*$ and normalize by the realized $S_{\mathrm{emp}} = \max_j(C_j - r_j)/p_j$, reporting $\rho_{S,\max} = S_{\max}/S_{\mathrm{emp}}$ together with $\rho_{S,99}$ and $\rho_{S,\mathrm{med}}$.

## 4 RESULTS AND DISCUSSION

Table 3 summarizes prediction quality. LGBM-Meta achieves the strongest overall accuracy: 51.2% Cov@25, 70.6% Cov@50, and $\rho$=0.912 through meta-stacking, while Two-Stage attains the highest recurring-job coverage (78.0% Cov@50) via classification-first routing. The consistent advantage over baselines confirms that task-signature features and empirical Bayes shrinkage are essential. Recurring jobs, which constitute 81.3% of the test set, exhibit markedly higher accuracy across all methods, validating the effective utility of historical job execution data. As shown in Table 4, each scheduling objective interacts with prediction quality differently. For total completion time, Meta-SPJF achieves a ratio of 1.106, a 10.5% degradation from clairvoyant SJF, and consistently outperforms PRR ($\lambda$=0.7), indicating that fully committing to predicted rankings supersedes hedging with round-robin. Makespan shows moderate sensitivity: Iso-LPPT (1.574) and CQR-SPPT (1.701) achieve the best ratios, as this objective depends primarily on identifying the largest jobs. Max-stretch reveals a qualitatively different ranking. SPRPT consistently dominates EDF-P, with Meta-SPRPT achieving an $11\times$ improvement over LAS/FB. EDF-P improves median stretch ($\rho_{\mathrm{med}} \approx 4.5$ for Meta-EDF, a $40\times$ gain over LAS/FB), yet its worst case remains comparable to LAS/FB. HRAS drives EDF-P to $\rho_{\max} \approx 1700$, over $8\times$ worse than Meta-EDF, while SPRPT degrades only from 18 to 51, confirming that deadline-based scheduling consistently amplifies absolute size errors.

Table 3: LASched prediction performance on PAI trace. Cov@$k$ = fraction of predictions within $k\%$ relative error.

| | Method | Task Signatures | Causal History | Ensemble Type | Cov@25 (%) All / Rec. | Cov@50 (%) All / Rec. | RMSLE (All) | $\rho$ (All) |
|---|---|---|---|---|---|---|---|---|
| Baseline | Single-Stage | ✗ | Rolling | None | 19.5 / 28.5 | 36.2 / 51.6 | 1.568 | 0.690 |
| | CRR++ | ✗ | Rolling | Cluster | 20.2 / 29.2 | 37.7 / 53.1 | 1.621 | 0.665 |
| | Scale-Experts | ✗ | Rolling | Bucket | 19.6 / 28.0 | 38.0 / 53.3 | 1.611 | 0.673 |
| | Recency-Fixed | ✗ | Rolling | Temporal | 20.2 / 28.2 | 36.9 / 50.9 | 1.578 | 0.685 |
| Advanced | CQR-Stack | ✓ | EWM+EB | Conformal | 49.0 / 57.1 | 67.5 / 76.8 | 0.944 | 0.897 |
| | HRAS | ✓ | – | None | 25.6 / 29.1 | 46.9 / 52.5 | 1.373 | 0.801 |
| | QMed+Iso | ✓ | EWM+EB | Isotonic | 45.9 / 53.4 | 67.0 / 76.2 | 0.975 | 0.888 |
| | **LGBM-Meta** | ✓ | EWM+EB | Meta-stack | **51.2** / **58.3** | **70.6** / 77.5 | **0.871** | **0.912** |
| | Two-Stage | ✓ | EWM+EB | Classify | 49.8 / 58.0 | 68.8 / **78.0** | 0.945 | 0.897 |
| | Weighted-Rec | ✓ | EWM+EB | Temporal | 39.9 / 45.5 | 63.2 / 70.9 | 0.948 | 0.896 |

Notes: Rec. = task signature seen in training. Baselines: 52.0% recurring (no signature features); Advanced: 81.3% recurring (with signature features and EB shrinkage).

Table 4: Complete scheduling performance across three objectives; lower is better. Max-stretch on $n$=5,000 jobs; total completion time on $n$=10,000 jobs; makespan ratios averaged over $m \in \{5, 10, 20, 50, 100\}$.

| (A) Max-Stretch | | | | (B) Total Compl. Time | | (C) Makespan | |
|---|---|---|---|---|---|---|---|
| Algorithm | $\rho_{S,\max}$ | $\rho_{S,99}$ | $\rho_{S,\mathrm{med}}$ | Algorithm | Ratio | Algorithm | Ratio |
| OPT (EDF at $S^*$) | **1.000** | **1.000** | **1.000** | SRPT | **1.000** | LPT | **1.000** |
| SRPT | 1.581 | 1.304 | 0.932 | SJF | 1.001 | SPT | 1.539 |
| LAS/FB | 205.28 | 134.18 | 187.31 | RR | 1.975 | Random | 1.898 |
| | | | | FIFO | 5.372 | | |
| CQR-SPRPT | 21.45 | 5.802 | 6.219 | CQR-SPJF | 1.135 | CQR-LPPT | 1.721 |
| CQR-EDF | 400.06 | 75.78 | 13.30 | CQR-PRR | 1.328 | CQR-SPPT | **1.701** |
| HRAS-SPRPT | 51.10 | 7.517 | 14.05 | HRAS-SPJF | 1.686 | HRAS-LPPT | 1.937 |
| HRAS-EDF | 1700.5 | 514.83 | 405.33 | HRAS-PRR | 1.887 | HRAS-SPPT | 1.753 |
| Iso-SPRPT | 21.66 | 5.162 | 6.327 | Iso-SPJF | 1.161 | Iso-LPPT | **1.574** |
| Iso-EDF | 400.36 | 86.11 | 22.63 | Iso-PRR | 1.354 | Iso-SPPT | 1.717 |
| Meta-SPRPT | **18.13** | 4.643 | **4.718** | Meta-SPJF | **1.106** | Meta-LPPT | 1.689 |
| Meta-EDF | **204.66** | **34.13** | **4.504** | Meta-PRR | **1.292** | Meta-SPPT | 1.731 |
| TwoSt-SPRPT | 20.77 | **4.469** | 5.469 | TwoSt-SPJF | 1.124 | TwoSt-LPPT | 1.643 |
| TwoSt-EDF | 411.24 | 88.62 | 9.436 | TwoSt-PRR | 1.310 | TwoSt-SPPT | 1.722 |
| Rec-SPRPT | 31.11 | 5.571 | 8.471 | Rec-SPJF | 1.172 | Rec-LPPT | 1.842 |
| Rec-EDF | 572.26 | 136.03 | 77.85 | Rec-PRR | 1.369 | Rec-SPPT | 1.943 |

## 5 CONCLUSION

We introduce **ATLAS**, a research-ready dataset comprising over 730k+ cluster jobs with comprehensive features and ground-truth sizes, alongside **LASched**, a standardized benchmark and implementation framework for learning-augmented scheduling. We establish rigorous prediction baselines evaluated via coverage, RMSLE, and rank correlation, and benchmark a suite of scheduling algorithms. Serving as a community reference point, our empirical evaluation demonstrates that systems utilizing calibrated predictors achieve near-optimal total completion times and competitive makespan ratios. For tail-sensitive objectives, algorithmic structure is as vital as predictive accuracy: while EDF with predicted deadlines substantially outperforms non-clairvoyant baselines, the worst-case stretch remains an order of magnitude above optimal, exposing a fundamental gap between aggregate fidelity and worst-case guarantees. The findings motivate key directions: (1) asymmetric or tail-aware loss functions to explicitly align model training with scheduling metrics, (2) distributionally robust optimization (e.g., CVaR) to minimize worst-case stretch, and (3) the co-design of prediction and deadline assignment mechanisms to fully exploit the structural advantages of EDF-style policies. Future work will extend the cross-dataset validation and optimization objectives.

ACKNOWLEDGMENTS

Zhiyun Jiang gratefully acknowledges the support of her husband, Si Wu. Professor Albert Zomaya and Dr Tianming Zhao would like to acknowledge the support of the Australian Research Council Research Hub for Future Digital Manufacturing (IH230100013).

**Reproducibility Statement.** We make our work reproducible along three aspects: data, prediction, and scheduling. The clear step-by-step user guide, including data downloading notes for different systems, code scripts running suggestions, and a detailed evaluation method, is in anonymized link, referred to readme.md file in the following link: `https://github.com/zhiyunjiang0810/non-clairvoyant-with-predictions`. Three reproducible information are listed.

(i) *Data.* ATLAS is derived from the public Alibaba PAI–2020 trace with a novel formalized job schema and label construction; we release an anonymized repository, shown in abstract, with scripts to rebuild the submit-time job table and ground-truth labels from raw datasets, including checks for terminated rows only and exact time semantics (earliest task start, latest task end). Please see dataset link in the abstract and Section 2: Dataset. Users can employ this dataset to construct interested columns, such as maximum task duration and instance duration. Also, users can make Python plots to see job size distribution, actual resource utilization rate, which could be both at submit-time or post-execution, any workload characterization preferred.

(ii) *Prediction.* We release code to reproduce the split (70%/15%/15% by submit time), train-only feature engineering (resources, temporal signals, recurrence signatures, strictly causal group/user histories with `shift(1)`, and label encoding), and all six calibrated baselines with validation-only calibration; configuration files and fixed random seeds are provided to regenerate Table 3 end-to-end (Cov@25/50, RMSLE, and Spearman's $\rho$). Users can use traditional ML models to make job size and task size predictions, which is also provided, and technical details are shown in Appendix C.

(iii) *Scheduling.* The benchmark includes an executable simulator with reference implementations of all policies and the exact normalizations used in Table 4: $\sum_j C_j$ reported relative to SRPT; makespan reported relative to the preemptive lower bound $\text{OPT}_{\text{pre}}$; and max-stretch computed via an EDF-feasibility test at the bisection optimum. Scripts are provided to recreate every number from a fixed commit.

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

# A NOTATIONS

## A.1 BASIC ENTITIES AND TIME VARIABLES

| Symbol | Name | Meaning | Example |
|--------|------|---------|---------|
| $J_j$ | Job | A job submitted by a user | Training job for classifier |
| $t$ | Task | A role within a job | Worker task in distributed training |
| $i$ | Instance | One copy of a task running | Worker instance #3 out of 10 |
| $m$ | Machine | Physical server in cluster | Server with 8 V100 GPUs |
| $r_j$ | Arrival time | When job was submitted | Submitted at 10:00 AM |
| $\tau$ | Time variable | Any point in time | Checking availability at 10:15 AM |
| $s_t$ | Task start time | When task begins running | Task starts at 10:30 AM |
| $s_j$ | Job start time | When first task starts | Job starts with earliest task |
| $q_j$ | Queuing delay | Time spent waiting | Waited 30 minutes for resources |

## A.2 SETS AND COLLECTIONS

| Symbol | Name | Meaning | Example |
|--------|------|---------|---------|
| $T(j)$ | Task set | All tasks belonging to job $j$ | {PS, Worker, Evaluator} |
| $I(t)$ | Instance set | All instances of task $t$ | {Worker-1, ..., Worker-10} |
| $\Gamma_t$ | GPU type set | Compatible GPU types | {V100, P100} but not T4 |

## A.3 RESOURCE VECTORS AND DEMANDS

| Symbol | Name | Meaning | Example |
|--------|------|---------|---------|
| $\mathbf{r}_t$ | Resource request | Resources per instance | [2 GPUs, 16 CPUs, 64GB RAM] |
| $\mathbf{c}_m$ | Machine capacity | Total machine resources | [8 GPUs, 96 CPUs, 512GB RAM] |
| $n_t$ | Instance count | Number of task copies | 10 worker instances |

## A.4 RESOURCE UTILIZATION

## A.5 JOB PROCESSING TIME CALCULATION

We define job processing time based on a hierarchical Fork-Join model.

**1. Instance Duration** For an instance $i$ running on the interval $[s_i, e_i)$:

$$d_i = e_i - s_i \tag{1}$$

**2. Task Span (Fork-Join for Instances)** A task $t$ completes only when its last instance finishes.

$$s_t = \min_{i \in I(t)} s_i, \quad e_t = \max_{i \in I(t)} e_i \tag{2}$$

$$d_t = e_t - s_t = \max_{i \in I(t)} e_i - \min_{i \in I(t)} s_i \tag{3}$$

*Note: For gang-scheduled tasks ($g_t = 1$), $s_i = s_t$ for all $i$, so $d_t = \max_i d_i$.*

**3. Job Processing Time (Fork-Join for Tasks)** A job $J_j$ completes only when its last task finishes.

$$S = \min_{t \in T(j)} s_t, \quad E = \max_{t \in T(j)} e_t \tag{4}$$

$$p_j^* = E - S \tag{5}$$

*This definition correctly captures the true resource occupancy window, including barriers from staggered task starts, unlike mean-based aggregations.*

| Symbol | Name | Meaning | Example |
|--------|------|---------|---------|
| $\mathbf{R}_t$ | Total task request | $\mathbf{R}_t = n_t \cdot \mathbf{r}_t$ | Total resources for all workers |
| $\mathbf{R}_j$ | Total job request | $\mathbf{R}_j = \sum_{t \in T(j)} \mathbf{R}_t$ | Sum of all tasks' requests |
| $u_{i,k}$ | Instance utilization | Average usage of resource $k$ by instance $i$ | 85% GPU usage over execution |
| $A_{i,k}$ | Resource-time | $A_{i,k} = u_{i,k} \cdot c_{m(i),k} \cdot d_i$ | Total GPU-seconds consumed |

## A.6 KEY SCHEDULING CONSTRAINTS

**1. Resource Capacity Constraint**

$$\sum_i x_{i,m}(\tau) \cdot \mathbf{r}_{t(i)} \leq \mathbf{c}_m \tag{6}$$

*Total resources used by all instances on a machine cannot exceed that machine's capacity*

**2. Gang Scheduling Constraint** (when $g_t = 1$)

$$\sum_m \sum_{i \in I(t)} x_{i,m}(\tau) = n_t \tag{7}$$

*All $n_t$ instances of the task must be placed at the same time $\tau$*

**3. Locality Constraint** (when $\ell_t = 1$)

$$\sum_{i \in I(t)} x_{i,m_t}(\tau) = n_t \tag{8}$$

*All instances must be on the same machine $m_t$*

**4. GPU Type Constraint**

$$x_{i,m}(\tau) = 0 \quad \text{when } g(m) \notin \Gamma_t \tag{9}$$

*Cannot place instances on machines with incompatible GPU types*

## B DATASET CHARACTERISTICS AND JOB EXAMPLES

To illustrate the diversity of workloads in the ATLAS dataset and clarify the nature of the Processing Time label, Table 5 presents three representative jobs drawn directly from the trace. These examples showcase different scales of operation, from small inference tasks to large-scale distributed training.

Crucially, the workloads captured in the Alibaba PAI trace are non-preemptible. Once a job begins execution, it runs to completion without interruption by the scheduler. Therefore, the Processing Time reported in the final column of Table 5 represents the actual, continuous duration of the job from start to finish. This single value accurately reflects the job's size for prediction tasks, as there are no preemption or resumption dynamics to model. The table highlights key features used for prediction, such as the number of tasks and instances, requested resources (CPU, GPU, memory), and workload type. The wide range of processing times, from just over two minutes to more than 12 hours, demonstrates the challenge of the prediction task.

Table 5: Three representative job examples from the ATLAS submit-time dataset, illustrating different job types, scales, and their corresponding processing times. The Processing Time serves as the prediction label and represents the uninterrupted execution duration, as jobs in this trace are not preempted.

| Job Type | pai_job_table | | | pai_task_table | | | | | pai_group_tag_table | | | Processing Time |
|----------|------|------|------|------|------|------|------|------|------|------|------|------|
| | job_name | user_id | submit_time | tasks | instances | cpu (%) | gpu (%) | mem (GB) | group_tag | workload | recurrence | (Label) |
| Small Inference | d7eb43b8... | 5b1345f0... | 09:23:15 | 1 | 1 | 600 | 25 | 29.3 | 6c0d75d7... | - | 47 | **136 s** |
| Distributed Training | 84afa920... | d4d51aca... | 10:45:30 | 2 | 26 | 15,100 | 625 | 52.0 | aba828a1... | ctr | 12 | **9,493 s** |
| Large Scale | e6145fb3... | df2899e2... | 14:12:45 | 2 | 105 | 55,000 | 4,000 | 2,050.8 | e9d4c564... | - | 3 | **44,632 s** |

## C    TRADITIONAL ML PREDICTION MODELS TRIED

Our preliminary methods explored four baselines: (1) a single-stage gradient-boosted trees model on the log target with monotone constraints on obvious scale features (instances, GPUs), to encode weak priors and reduce pathological splits in sparse regions Ke et al. (2017). (2) a CRR++ "cluster→route→refit" variant that forms train-only k-means families, learns a router, and applies per-family experts with a global fallback—useful when coarse workload types exist but are unlabeled Pedregosa et al. (2011). (3) scale-bucket experts using train-quantile edges on instance/GPU counts to fit per-bucket regressor, again with a global fallback; and (4) a simple recency ensemble mixing a full-history model with a recent-window model to hedge concept drift Gama et al. (2014). These baselines improved over naive single models but exposed core gaps: poor uncertainty calibration and tail handling (heavy-tailed durations), weak rank fidelity in some regimes, brittle behavior for signatures, and limited drift adaptation from fixed mixtures. So, bad performances motivated our current seven-method toolkit that adds calibrated quantile intervals and blending, explicit monotone-safe calibration (isotonic), diversity via stacking, a principled gated-experts split, stronger recency weighting with validation-based ensembling, and empirical-Bayes priors for sparse signatures—addressing calibration, ranking, sparsity, and drift more systematically than the four preliminaries could alone.

---

**Algorithm 2** Common preprocessing (used by M1–M4)

1: **Input:** raw tables (job, task, tag), submit time $r_j$, label $p_j^\star$, features $\mathbf{x}_j$
2: **Split by time:** sort by $r_j$; pick cut times $t_{\text{train}} < t_{\text{val}}$; define $\mathcal{D}_{tr} = \{j : r_j < t_{\text{train}}\}$, $\mathcal{D}_{va} = \{j : t_{\text{train}} \leq r_j < t_{\text{val}}\}$, $\mathcal{D}_{te} = \{j : r_j \geq t_{\text{val}}\}$.
3: **Core features (submit-time only):** logs of totals ($\log(1 + x)$ for CPU/GPU/MEM/instances/tasks), per-instance ratios, cyclic time (`hour`, `wday`, sin / cos of 24h and 168h).
4: **Causal histories (group/user):** on the *log target* $y_j = \log(1 + p_j^\star)$, compute within-group expanding means and counts with a one-step `shift(1)`; time since previous submit; small shifted rolling means; an empirical-Bayes (EB) group mean $\mu^{EB}$ using only $\mathcal{D}_{tr}$ to set the global prior $\mu_0$.
5: **Train-only encodings:** map `user`/`group`/`workload`/`gpu_type_spec` to integer codes using the vocabulary in $\mathcal{D}_{tr}$; unseen $\mapsto$ `OTHER`.
6: **Sanity:** replace $\pm\infty$ and NaNs with 0 for numeric features; **never** touch labels in $\mathcal{D}_{va}, \mathcal{D}_{te}$ beyond metrics.

7: **Return:** design matrices $X_{tr}, X_{va}, X_{te}$ and vectors $y_{tr} = \log(1 + p^\star)$, $y_{va}, y_{te} = p^\star$.

---

**Algorithm 3** M1 — Single-Stage Gradient Boosting (log target) with causal histories

1: **Input:** $X_{tr}, X_{va}, X_{te}$ from Alg. 2; $y_{tr}, y_{va}, y_{te}$
2: **(optional) Monotone constraints:** choose a feature subset $\mathcal{M}^+$ expected to be non-decreasing (e.g., $\log(1 + \text{instances})$, $\log(1 + \text{GPU})$) and pass a monotonicity vector to the booster.
3: Train a gradient-boosted trees regressor on $y_{tr}$ with early stopping on $(X_{va}, y_{va})$ (all log-domain).
4: **Predict:** $\hat{y}_{te} \leftarrow \text{model}(X_{te})$; return $\hat{p}_{te} = \exp(\hat{y}_{te}) - 1$.
5: **Metrics:** report Cov@25/50, RMSLE, MAE, and Spearman on $(\hat{p}_{te}, y_{te})$.

---

**Algorithm 4** M2 — CRR++ (Cluster → Route → Refit) with per-family experts

1: **Input:** $X_{tr}, X_{va}, X_{te}$; $y_{tr}, y_{va}, y_{te}$; clusters $K$; min-support $n_{\min}$
2: **Standardize (train-only):** fit a scaler on $X_{tr}$; transform to $Z_{tr}, Z_{va}, Z_{te}$.
3: **Unsupervised families (train-only):** fit $K$-means on $Z_{tr}$; obtain family IDs $f_{tr}$; assign $f_{va}, f_{te}$ by `predict`.
4: **Router (train-only):** train a multi-class classifier to map $Z \mapsto f$ using $(Z_{tr}, f_{tr})$.
5: **Fallback regressor:** train a global log-target booster on $(X_{tr}, y_{tr})$ with early stopping on $(X_{va}, y_{va})$.
6: **Per-family experts:** for each $k \in \{1, \ldots, K\}$ with $\#\{j \in \mathcal{D}_{tr} : f_{tr}(j) = k\} \geq n_{\min}$, train a log-target booster on the subset $\{j : f_{tr}(j) = k\}$; optionally use $(X_{va}[f_{va} = k], y_{va}[f_{va} = k])$ for early stopping.
7: **Predict:** for each test sample $x$, set $\hat{k} \leftarrow \text{router}(z)$; if expert $k = \hat{k}$ exists use it, else use the fallback; return $\hat{p} = \exp(\hat{y}) - 1$.
8: **Metrics:** as in Alg. 3.

---

**Algorithm 5** M3 — Scale-Bucket Experts (train-quantiles → per-bucket models)

---

1: **Input:** $X_{tr}, X_{va}, X_{te}$; $y_{tr}, y_{va}, y_{te}$; train-only features $a =$ instances, $b =$ GPU; min-bucket $n_{\min}$
2: **Train-only bucket edges:** compute quantiles $Q_a$ and $Q_b$ on $(a, b)$ over $\mathcal{D}_{tr}$ (e.g., $\{0.5, 0.9, 0.99\}$); define bucketizers $\text{bin}_a, \text{bin}_b$.
3: **Assign buckets:** $u_j = \max\{\text{bin}_a(a_j), \text{bin}_b(b_j)\}$ for all $j$ in train/val/test.
4: **Global fallback:** train a log-target booster on all of $\mathcal{D}_{tr}$ with early stopping on $\mathcal{D}_{va}$.
5: **Per-bucket experts:** for each bucket $u$ with $\#\{j \in \mathcal{D}_{tr} : u_j = u\} \geq n_{\min}$, train a log-target booster on $\{j : u_j = u\}$; optionally early-stop on $\{j \in \mathcal{D}_{va} : u_j = u\}$.
6: **Predict:** for each test sample with bucket $u$, use expert($u$) if available; else use the global fallback; return $\hat{p} = \exp(\hat{y}) - 1$.
7: **Metrics:** as in Alg. 3.

---

**Algorithm 6** M4 — Recency Ensemble (full vs. recent window; train-only gates)

---

1: **Input:** $X_{tr}, X_{va}, X_{te}$; $y_{tr}, y_{va}, y_{te}$; training times $\{r_j : j \in \mathcal{D}_{tr}\}$; window quantile $q$; mixture weight $\alpha$
2: **Full model:** train a log-target booster on $(X_{tr}, y_{tr})$; early-stop on $(X_{va}, y_{va})$.
3: **Recent cut (train-only):** set $t_q \leftarrow \text{Quantile}_q(\{r_j : j \in \mathcal{D}_{tr}\})$; define $\mathcal{D}_{tr}^{\text{recent}} = \{j \in \mathcal{D}_{tr} : r_j \geq t_q\}$.
4: **Recent model (if enough support):** train a log-target booster on $\mathcal{D}_{tr}^{\text{recent}}$, early-stopped on $(X_{va}, y_{va})$; otherwise skip.
5: **Predict & mix:** let $\tilde{p}^{(full)} = \exp(\hat{y}^{(full)}) - 1$, $\tilde{p}^{(rec)} = \exp(\hat{y}^{(rec)}) - 1$ (if present); output $\hat{p} = (1 - \alpha)\tilde{p}^{(full)} + \alpha\tilde{p}^{(rec)}$ if recent model exists, else $\tilde{p}^{(full)}$.
6: **Metrics:** as in Alg. 3.

---

## D    PREDICTION MODEL DESCRIPTION

We model $y_j = \log(1 + p_j^\star)$ from submit-time features $\mathbf{x}_j$ using gradient boosting with validation-based calibration. Our six methods address different challenges in job duration prediction:

**(1) Conformal Quantile Regression (CQR)** (Romano et al., 2019) addresses prediction uncertainty by learning the conditional distribution rather than just point estimates. We train three LightGBM models with quantile loss at $\alpha \in \{0.1, 0.5, 0.9\}$ to predict the 10th, 50th, and 90th percentiles of job duration. On the validation set, we compute the nonconformity score $r = \max(Q_{0.1} - y, \ y - Q_{0.9})$ for each example and obtain the conformal correction $k = \mathcal{Q}_{0.6}(\max(r, 0))$, which captures the typical prediction interval miscalibration. The calibrated bounds are $L = Q_{0.1} - k$ and $U = Q_{0.9} + k$, expanding the quantile interval to achieve the desired coverage. The final prediction blends $Q_{0.5}$, the calibrated bounds $[L, U]$, and contextual features like group means using Ridge regression. This approach is particularly effective for jobs with high uncertainty—for instance, experimental ML training jobs where duration depends on convergence criteria, or data processing jobs where input size varies significantly. The method provides both accurate point estimates and reliable confidence intervals.

**(2) Isotonic Calibration** (Zadrozny & Elkan, 2002; Kuleshov et al., 2018) corrects systematic prediction biases while preserving ranking order. The method fits a monotonic non-decreasing function $\phi : \mathbb{R} \to \mathbb{R}$ mapping raw predictions to calibrated values, ensuring $\phi(x_1) \leq \phi(x_2)$ whenever $x_1 \leq x_2$. This is crucial when the model consistently over-predicts short jobs (e.g., quick validation scripts that always take 5 seconds but are predicted as 30 seconds) or under-predicts long jobs (e.g., full model training that takes 10 hours but is predicted as 2 hours). The isotonic regression finds the optimal step function that minimizes squared error on the validation set while maintaining monotonicity. This property is essential for scheduling decisions where relative job ordering matters—if job A is predicted to be shorter than job B, this relationship is preserved after calibration.

**(3) Meta-Stacking** (Wolpert, 1992) leverages model diversity to improve robustness. We train four base models with different loss functions: (i) L2 loss for standard regression, (ii) Huber loss ($\delta = 0.9$) for robustness to outliers like crashed jobs or anomalously long runs, (iii) quantile loss for median prediction, and (iv) heavily regularized L2 ($\alpha = 0.5, \lambda = 1.0$) to prevent overfitting. Each model captures different aspects: L2 minimizes average error, Huber handles extreme cases, quantile focuses on the median behavior, and regularized models provide stable baselines. The meta-learner (another LGBM) takes these base predictions, their standard deviation (measuring disagreement), and contextual features (group history, signature statistics) as input. It learns non-linear

combinations—for example, trusting the Huber model more when base predictions diverge significantly (indicating potential outliers), or weighting the regularized model higher for users with limited history.

**(4) Two-Stage Gated Experts** (Jordan & Jacobs, 1994) recognizes that different job types require different prediction strategies. The gating network (classifier) first categorizes jobs into three duration classes based on the 30th and 70th percentiles of training durations: (i) *short jobs* ($<$30th percentile, typically $<$100s): quick validation runs, status checks, or small data samples; (ii) *medium jobs* (30th–70th percentile, 100s–1000s): regular training epochs, moderate data processing; (iii) *long jobs* ($>$70th percentile, $>$1000s): full model training, large-scale data processing, or hyperparameter sweeps. Each category gets a specialized expert model with appropriate complexity—simple models for predictable short jobs, complex deep trees for variable long jobs. The final prediction aggregates expert outputs weighted by soft probabilities: $\hat{y} = \sum_{c \in \{\text{short, med, long}\}} P(c|\mathbf{x}) \cdot E_c(\mathbf{x})$. This prevents short jobs from being influenced by patterns from marathon training runs and vice versa.

**(5) Weighted Recency** (Gama et al., 2014) adapts to temporal drift in workload patterns. Computing clusters exhibit temporal patterns: new framework releases change typical training times, approaching deadlines increase job submissions, and hardware upgrades affect processing speeds. We train three models on progressively recent data windows: (i) full history for stable long-term patterns, (ii) recent 50% for medium-term trends, (iii) recent 20% for immediate patterns. Each training sample receives weight $w_t = \exp(-\lambda(T - t))$ where $T$ is the current time and $\lambda$ controls decay rate. Models are combined using validation performance weights—if recent models show lower validation error, they receive higher weight in the ensemble, automatically adapting to drift. For example, after a PyTorch version upgrade that speeds up training by 30%, the recent-20% model would quickly adapt while the full-history model provides stability.

**(6) Historical Recency-Aware with Shrinkage (HRAS)** addresses the cold-start problem for rare job signatures. A signature combines user, group, workload type, and resource requirements (bucketed into deciles). Rare signatures include: (i) new users or groups with no history, (ii) unusual resource combinations (e.g., user `alice` from `vision_group` suddenly requesting 8 GPUs when historically using only CPUs), (iii) infrequent workload types (e.g., monthly financial reports in a primarily ML-focused cluster). For signature $s$ with $n_s$ historical observations and mean duration $\mu_s$, we apply Empirical Bayes shrinkage: $\bar{\mu}_s = (n_s\mu_s + \lambda\mu_0)/(n_s + \lambda)$, where $\mu_0$ is the global mean and $\lambda = 5$ is the shrinkage strength. When $n_s = 0$ (completely new pattern), the prediction equals the global mean. As $n_s$ grows, the prediction gradually shifts toward the signature-specific mean. The exponentially weighted component gives more weight to recent instances of the signature, capturing evolution in user behavior.

