# OpenReview forum: "ATLAS: Alibaba Dataset and Benchmark for Learning-Augmented Scheduling"
_ICLR.cc/2026/Conference — ICLR 2026 Poster_

### Official Review · Reviewer_shox · 2025-10-18

**Soundness:** 3
**Presentation:** 2
**Contribution:** 2
**Rating:** 6
**Confidence:** 3

**Summary:**

The paper presents a new real-world dataset for evaluating algorithms with predictions. In particular, the paper addresses non-clairvoyant scheduling, where job sizes are unknown to a scheduler. Instead, a scheduler has access to predictions about the job, for example, the job's processing times. Non-clairvoyant scheduling belongs to one of the most studied online problems within the area of algorithms with predictions. However, many results are only of theoretical nature, and it is unclear whether these algorithm bring a benefit in practice.

The paper proposes a dataset based on real world data of Alibaba's datacenter and preprocesses that for evaluating such algorithms. There are two main components:

1. The main component is the dataset itself. It is very diverse and comes with different labels.
2. The second component is a ready-to-use benchmark that already implements baselines such as simple learning-augmented algorithms and optimum solutions. It also comes with benchmark prediction models that can be used for generating e.g. job size predictions. Considered objectives are total completion time, maximum stretch and makespan

The paper gives characterizations of the dataset, a detailed explanation of the benchmark suite, and an evaluation of multiple learning-augmented and classic algorithms.

**Strengths:**

- To the best of my knowledge, this is the first attempt to introduce a benchmark for learning-augmented scheduling algorithms, which seems to be overdue. In the past, there often has been critique that learning-augmented algorithms are evaluated on synthetic data. Thus, I think this benchmark suite is a good contribution for the field and creates a bridge to practical applications.
- The benchmark considers different objectives to capture a large variety of problems.
- The dataset and benchmark are well-structured: a proper feature engineering pipeline, empirical-Bayes shrinkage, and conformal calibration.
- The authors show attention to reproducibility.

**Weaknesses:**

- The introduction is slightly dense, it assumes that readers are familiar with learning-augmented algorithms; also the statements about the different algorithm are a bit confusing in my opinion. I would like to see a revised introduction in the camera-ready version.
- One could argue that the contribution of a dataset is not sufficient for a ML theory conference. However, since benchmarks are rather novel for learning-augmented algorithms, I would not value this too much.

I would be happy to see the paper accepted, but would not fight for it.

**Questions:**

- L25 mutli-stage?
- L33: whitespace missing
- L35: "non-clairvoyant scheduling has a larger total completion time" what do you mean here?
- L105 "ALTAS" -> "ATLAS"

---

> ### Author Response · Authors · 2025-11-21
>
> We sincerely thank the reviewer for their positive assessment of our benchmark’s rigor and for recognizing that a standardized benchmark for learning-augmented scheduling is overdue. We are highly encouraged that you value our effort in bridging the gap between theoretical assumptions and practical system realities. We particularly appreciate your constructive feedback regarding the presentation density; clarity is essential for a benchmark paper, and your comments have helped us refine the presentation of our work for the broader ML community. We address your specific questions below.
>
> **Questions**
>
> **1. Clarification of "Non-Clairvoyant" Performance (L35)**
>
> We intended to convey that non-clairvoyant algorithms are strictly suboptimal compared to clairvoyant ones because they lack job size information. In the revision, we have rewritten this sentence to be precise.
>
> > *As job sizes are unknown at arrival, the scheduler cannot implement an optimal clairvoyant strategy such as SRPT for total completion time; consequently, non-clairvoyant algorithms achieve suboptimal scheduling performance (Motwani et al., 1994).*
>
> **2. Minor corrections**
> Thank you for catching these typos.
>
> * **L25 & L33:** Fixed.
> * **L105:** Corrected "ALTAS" to "ATLAS".
>
> **W1. Presentation and introduction density**
>
> Thanks for your kind advice that our introduction is slightly dense, and it might be hard to read for a broader audience. We think this is valuable feedback, and we have updated the introduction to bring greater impact to the whole ML community.
>
> For paragraph 2 in the introduction, we improved the flow of mathematical definitions to make the introduction less dense:
>
> > *To illustrate the effect of predictions, consider the single‑machine scheduling to minimize total completion time $\sum C_j$, where $C_j$ is the completion time of job $J_j$, in Figure 1. Suppose we have four jobs released at $r_j$ with unknown sizes $p_j^{\star}$. Without job predictions, First-In-First-Out (FIFO) runs jobs in arrival order, yielding a total completion time of 51.*
>
> In paragraph 3, we smooth the transition from theoretical works to real-world challenges:
>
> > *However, real production clusters often violate the core assumptions underlying these theoretical models. In practice, jobs execute as multi-step workflows (e.g., preprocessing before training) where early-stage failures can terminate the sequence.*
>
> For the **Issues with current datasets and benchmarks** paragraph in the introduction, we refined the whole paragraph for clarity. The changes are highlighted in the updated file to make them easier to track:
>
> > *First, existing production traces offer limited data for training and evaluating predictors for job processing times. Google's Borg traces (Tirmazi et al., 2020) normalize processing times and obfuscate job identities, removing rich context like user patterns, job types, resource requests, and historical behavior. Azure public datasets (Cortez et al., 2017) and Microsoft’s Virtual Machine (VM) allocation traces (Lu et al., 2017) focus primarily on VM provisioning, exposing utilization rates while omitting job structures or exact completion times. The Alibaba trace (Weng et al., 2022) provides job structures but was designed for workload characterization rather than scheduling evaluations.*
> >
> > *Second, most theoretical studies rely exclusively on synthetic workloads (Zhao et al., 2022; Benomar et al., 2024), limiting job sizes to standard exponential, Pareto, or uniform distributions that miss the complex patterns found in real systems.*
> >
> > *Third, the field lacks a standardized evaluation benchmark: a clear, reproducible specification of (a) the scheduling framework (online/offline, (non‑)preemptive, number of machines), (b) how predictors are trained and validated, and (c) how results are reported and normalized. Consequently, different studies adopt incompatible problem formulations, metrics, and experimental setups, such as work by Fan et al. (2022), Im et al. (2023), and Bampis et al. (2023), making cross-paper algorithm comparisons difficult. Furthermore, many overlook temporal constraints (training on past, testing on future), failing to restrict features to historical information, or skip calibration–test separation, risking information leakage that violates non‑clairvoyant assumptions (Kapoor et al., 2023).*

---

> ### Author Response · Authors · 2025-11-21
>
> **W1. Clear scheduling algorithms**
>
> We have changed the **Scheduling algorithm** paragraph and refined it into more logical paragraphs, shown below.
>
> > **Baseline Algorithms.**
> > For non-clairvoyant baselines, we use FIFO as the standard online default (Weng et al., 2022), RR, which shares processor capacity equally among active jobs (Motwani et al., 1994), and LAS (Least-Attained-Service), which prioritizes the job that has received the least service so far (Nuyens et al., 2008).
> > For clairvoyant baselines, which serve as offline performance bounds, we employ SRPT, the optimal policy for minimizing total completion time (Schrage, 1968), and SJF (Shortest Job First).
> > For the multi-machine makespan objective, we evaluate LPT (Longest Processing Time), which balances load by assigning the largest job to the least-loaded machine (Della Croce et al., 2020), alongside SPT (Shortest Processing Time) and a Random assignment baseline.
>
>
>
> > **Scheduling Algorithms.**
> > These algorithms integrate predictions $\hat{p}_j$ generated by our prediction benchmark models (e.g., CQR, TwoSt) into online decision-making.
> > For **total completion time**, we evaluate SPJF (Mitzenmacher, 2019) and PRR (Preferential Round-Robin). PRR is a robust mechanism that reserves a processor share $\lambda$ for the job with the smallest $\hat{p}_j$ while distributing the remaining rate $(1-\lambda)$ equally among all jobs (Kumar et al., 2018).
>
> > For **max-stretch**, we evaluate SPRPT (SRPT using predicted remaining work) and EDF-P, an Earliest-Deadline-First policy that schedules based on predicted deadlines $d_j = r_j + S_{adv} \cdot \hat{p}_j$.
> > For **makespan**, we substitute true sizes with predictions to create LPPT (Longest Predicted Processing Time) and SPPT (Shortest Predicted Processing Time), a prediction variant of SPT, evaluating how prediction errors impact scheduling policies.

---

> > ### Comment · Reviewer_shox · 2025-11-25
> >
> > I thank the authors for their detailed response and for showing their willingness in improving the presentation.

---

### Official Review · Reviewer_tfJj · 2025-10-22

**Soundness:** 3
**Presentation:** 4
**Contribution:** 3
**Rating:** 8
**Confidence:** 3

**Summary:**

This paper introduces ATLAS, a large-scale dataset of over 730,000 jobs from Alibaba's PAI production cluster, which addresses the lack of public scheduling datasets containing ground-truth processing times. The dataset is specifically designed for non-clairvoyant scheduling research by strictly using only features available at submission time, and the authors build the LASched benchmark on top of it. LASched provides implementations and evaluations for both job size prediction tasks (using baselines like CQR and Meta-stacking) and scheduling tasks (using algorithms like SPJF and LPPT), serving as a reproducible foundation for comparing algorithms on real-world workloads.

**Strengths:**

* The paper convincingly argues that existing datasets for scheduling research are insufficient, as they either lack ground-truth processing times or are not publicly available. ATLAS directly fills this gap with a large-scale, public trace from a real-world production system.

* The dataset provides complete ground-truth job sizes, a critical feature missing from other traces. It is rigorously prepared as a non-clairvoyant dataset, strictly excluding post-execution metrics to prevent information leakage and ensure it is research-ready.

* The authors provide LASched, a full benchmark built on ATLAS that defines both prediction and scheduling tasks. This includes implementations and evaluations for multiple prediction models and scheduling algorithms , establishing a strong baseline for future work.

**Weaknesses:**

* Maybe I'm wrong but there seems to be a discrepancy between the complexity of the data source and the simplicity of the scheduling benchmark. The ATLAS dataset is sourced from a large scale, heterogeneous production environment with over 1,800 machines and complex constraints like gang scheduling. However, the LASched benchmark evaluates key metrics, such as total completion time and max-stretch, using a simplified single machine setting. This simplification might obscure the real world impact of prediction errors, which could be amplified in a complex, multi machine environment with heterogeneous resources and stricter constraints.

* The dataset's two month time span limits its ability to capture long term temporal dynamics. While the trace includes over 730,000 jobs and exhibits clear patterns, a two month window is likely insufficient to model longer cycles, such as quarterly workload peaks or concept drift caused by major software framework updates. The paper does propose models to address drift, but the dataset itself may not provide enough data for researchers to fully validate the robustness of scheduling algorithms against these important long term workload evolutions.

**Questions:**

* Since some metrics are measured on a single machine, how do you envision the benchmark evolving to incorporate real-world complexities, and what new challenges might arise when evaluating learning-augmented algorithms in that more realistic setting?

* Do you have plans to release an extended version of ATLAS, perhaps covering a full year, or how would you recommend researchers use this dataset to develop models that are robust to such long-term changes?

---

> ### Author Response · Authors · 2025-11-21
>
> We sincerely thank the reviewer for the encouraging positive assessment and for recognizing ATLAS as a rigorously prepared and convincing contribution. We particularly appreciate your acknowledgment of our effort to build a complete benchmark ecosystem. We address your insightful comments regarding complexity and future plans below.
>
> **Q1. Benchmark complexity vs. Real-world complexity.**
>
> We appreciate this observation. We wish to clarify that LASched does not reduce all problems to single-machine settings; rather, we selected the specific theoretical model (single vs. parallel) for each objective that allows for **provably optimal baselines**, enabling rigorous performance ratios.
>
> **1. Total completion time ($1|r_j,\text{pmtn}|\sum C_j$)**
> We evaluate total completion time on a single machine with online arrivals and preemption. This is the standard model [1] where **SRPT (Shortest Remaining Processing Time)** is provably optimal, as established by Schrage [2].
> * **Technical Rationale:** We utilize this single-machine model because it is the *minimal complexity setting* where optimal baselines exist. Learning-augmented algorithms like SPJF and PRR [3] are analyzed in this specific context.
> * **Hardness Constraint:** Extending this to parallel machines ($P|r_j,\text{pmtn}|\sum C_j$) renders the problem **NP-hard** even with known processing times [4]. Consequently, computing a ground-truth OPT for benchmarking becomes computationally intensive.
>
> **2. Makespan ($P||C_{\max}$)**
> We strictly evaluate Makespan on **$m$ parallel identical machines** with batch release.
> * **Technical Detail:** We vary $m \in \{5, 20, 50, 100\}$ to capture load-balancing dynamics. This corresponds to the standard theoretical setting for classical algorithms like **LPT (Longest Processing Time)** [5], and prediction-augmented variants like LPPT.
> * **Rationale:** Makespan minimization is trivial on a single machine. It is only meaningful when the load can be distributed across multiple processors.
>
> **3. Max-stretch ($1|r_j,\text{pmtn}|S_{\max}$)**
> We evaluate max-stretch ($S_{\max} = \max_j \frac{C_j - r_j}{p_j}$) on a single machine [6].
> * **Technical Rationale:** We rely on the **EDF (Earliest Deadline First)** feasibility framework. On a single processor, EDF provides an exact feasibility characterization [7], allowing us to compute the exact Optimal value via binary search on the max-stretch value.
> * **Comparison to Multi-machine:** In parallel machine settings, the literature typically studies *average* stretch or flow time due to the difficulty of minimizing worst-case objectives without migration assumptions [8]. Our design prioritizes exact optimality checks to validate prediction robustness.
>
> **4. Benchmark design**
>
> This design reflects established scheduling theory: different objectives are studied in different machine models because that is where clean theoretical baselines and performance guarantees exist. The key contribution of ATLAS is providing **realistic job distributions**: ground-truth sizes, recurrence patterns, resource requests, and arrival times, used to evaluate any theoretical scheduling model. Researchers studying gang scheduling can use ATLAS jobs in gang-scheduling simulators; those studying heterogeneous machines can leverage our GPU-type and resource metadata; those studying consistency and robustness can test their algorithm performances on our benchmark. The dataset variety supports more than these use cases, so we simply provide baseline implementations for the most theoretically grounded settings. So, assumptions about multiple-machine and more complex constraints can be simulated by any future researchers.
>
> **4. Challenges and future work**
>
> Extending to more complex models is indeed an exciting direction, and ATLAS is designed to support this evolution. The challenge is that such extensions often lack clean optimal baselines, making it difficult to rigorously quantify learning augmented algorithm performance, such as empirical competitive ratio. We view our current dataset and benchmark as establishing a foundation with well-understood theoretical properties, upon which the community can build more complex evaluations.

---

> ### Author Response · Authors · 2025-11-21
>
> **Q2. Extensions and recommendations**
>
> **1. Plans for extension**
>
> We are actively working to assess the feasibility of extending ATLAS to cover a full calendar year. The primary constraints involves the rigorous process of data anonymization and privacy review at scale, as well as the significant computational cost required to regenerate ground-truth labels for additional months. We aim to release an extended version if these constraints can be addressed, and we will announce any updates through the ATLAS repository.
>
> **2. Recommended usage for robustness**
>
> We believe the ATLAS dataset provides sufficient statistical depth for most research studies in scheduling with predictions. Spanning two months of continuous production workload with over 730,000 jobs, the dataset enables analyses that extend well beyond simple week-level patterns. Furthermore, despite the specific temporal window, ATLAS provides a rich multi-level hierarchy—including user, group, and workload tags, alongside granular CPU/GPU/memory plans—which supports realistic non-clairvoyant prediction and scheduling studies.
>
> For robustness, we highly recommend cross-dataset evaluation. Among widely used production ML scheduling traces, Philly provide 2-month-scale views [9]; Google’s public traces are month-scale as well, while Helios offers six months [10]. Researchers should note that given the substantial variation in the scope of each dataset, extensive data preprocessing is imperative. The Google research data can be checked using the link below.
>
> Google Cluster Data: <https://github.com/google/cluster-data>
>
> **References**
>
> [1] Ronald Lewis Graham, Eugene Leighton Lawler, Jan Karel Lenstra, and AHG Rinnooy Kan. Optimization and approximation in deterministic sequencing and scheduling: a survey. In Annals of discrete mathematics, volume 5, pp. 287–326. Elsevier, 1979.
>
> [2] Linus Schrage. A proof of the optimality of the shortest remaining processing time discipline. Operations Research, 16(3):687–690, 1968.
>
> [3] Ravi Kumar, Manish Purohit, and Zoya Svitkina. Improving online algorithms via ml predictions. In Proceedings of the 32nd International Conference on Neural Information Processing Systems, pp. 9684–9693, 2018.
>
> [4] Odile Bellenguez-Morineau, Marek Chrobak, Christoph Dürr, and Damien Prot. A note on np-hardness of preemptive mean flow-time scheduling for parallel machines. Journal of Scheduling, 18(3):299–304, 2015.
>
> [5] Ronald L. Graham. Bounds on multiprocessing timing anomalies. SIAM journal on Applied Mathematics, 17(2):416–429, 1969.
>
> [6] Michael A Bender, Soumen Chakrabarti, and Sambavi Muthukrishnan. Flow and stretch metrics for scheduling continuous job streams. In SODA, volume 98, pp. 270–279, 1998.
>
> [7] Chung Laung Liu and James W Layland. Scheduling algorithms for multiprogramming in a hard-real-time environment. Journal of the ACM (JACM), 20(1):46–61, 1973.
>
> [8] Luca Becchetti, Stefano Leonardi, and S Muthukrishnan. Scheduling to minimize average stretch without migration. In Proceedings of the eleventh annual ACM-SIAM symposium on Discrete algorithms, pp. 548–557, 2000.
>
> [9] Myeongjae Jeon, Shivaram Venkataraman, Amar Phanishayee, Junjie Qian, Wencong Xiao, and Fan Yang. Analysis of Large-Scale Multi-Tenant GPU clusters for DNN training workloads. In 2019 USENIX Annual Technical Conference (USENIX ATC 19), pp. 947–960, 2019.
>
> [10] Qinghao Hu, Peng Sun, Shengen Yan, Yonggang Wen, and Tianwei Zhang. Characterization and prediction of deep learning workloads in large-scale gpu datacenters. In Proceedings of the International Conference for High Performance Computing, Networking, Storage and Analysis, pp. 1–15, 2021.

---

### Official Review · Reviewer_ufpH · 2025-10-25

**Soundness:** 3
**Presentation:** 2
**Contribution:** 3
**Rating:** 8
**Confidence:** 4

**Summary:**

This paper publishes a “plug-n-play” data set for training job size predictors in the context of non-clairvoyant scheduling, a practically-relevant problem that has been extensively studied in the literature on learning-augmented algorithms.  The authors create ATLAS, a data set of real workloads distilled from Alibaba’s PAI trace.  Furthermore, they provide a benchmark environment called LASched that provides a common environment for evaluating learning-augmented schedulers, and give example prediction methods that perform well in this environment.  The paper is released alongside an anonymized repo that contains the data set, simulator code, and code for the example ML models.

**Strengths:**

The underlying problem (non-clairvoyant scheduling with job size predictions) is highly relevant in practice.   Providing a benchmark tool (LASched) alongside the data set will be appreciated by the community working on this problem as a way to speed up evaluation of new techniques and establish a common environment in which proposed scheduling algorithms can be directly compared.

In developing their own models for predicting job sizes, the authors additionally develop a feature engineering schema that is “ready to go” for other researchers developing new model architectures for the job size prediction task — they also show that these engineered features are effective for the ATLAS data set by proposing and evaluating their own predictors that perform well.

**Weaknesses:**

The Alibaba PAI data set that provides all of the underlying data for ATLAS has been publicly available since 2022 — ATLAS is simply a sensible cleaned and processed version of this data set.  This weakness is a mild one since the paper also provides a new benchmark and simulator environment that is of equal value to the community working on this problem.

While not necessarily expected from a paper in the datasets track, the example models trained using the data set are only evaluated with a fixed set of features and parameters — it would be good to see, for example, an ablation study of including/excluding certain features.

**Questions:**

Can you speak more to the decision to provide the predictor with information about the user and group?  I would typically expect that including these features would bias the model towards overfitting to the data in a way that may not be generalizable (e.g., to a different scheduling context with similar workloads but completely different users).  How many users/groups exist in the underlying trace, and are these literally tied to individual human beings, or are they tied to frameworks such as TensorFlow?

---

> ### Author Response · Authors · 2025-11-21
>
> We appreciate the reviewer’s positive assessment and their recognition of the practical relevance of non-clairvoyant scheduling. We are glad that the provided benchmark (LASched) and feature engineering pipeline were found to be valuable contributions beyond the raw dataset itself. Regarding your specific questions on feature selection (User/Group IDs) and generalizability, we provide the following clarifications to explain.
>
> ***Q1. Question on User and Group Features***
>
> **1. Clarifications of users and groups in ATLAS**
>
> In the Alibaba PAI trace, users represent anonymized submitter accounts. Based on the cluster's multi-tenant MLaaS platform design, these are primarily organizational accounts or service accounts tied to teams or projects rather than individual human beings. As shown in our paper Table 1 (Page 2), there are only 1,314 unique users submitting more than 730,000 jobs. These represent stable, long-term entities with consistent resource quotas and behavior patterns, making them a highly reliable feature for prediction.
>
> Group tags identify recurring workload patterns. Rather than being manually assigned, a group tag is automatically computed by hashing the job's configuration: the entry script, command-line parameters, input data paths, and output destinations. **Jobs with identical configurations receive identical group tags, even if submitted by different users at different times.** For example, a team's daily model retraining job, with the same parameters on the same dataset every morning, receives the same group tag each time it runs. The recurring nature of the workload is independent of which team member performs it.
>
> **2. Ablation study**
>
> We validate our 33-feature model through an ablation study on the test set, systematically adding feature groups to measure their contribution. Starting from a minimal baseline of 14 resource and temporal features, we progressively incorporate group-level features (workload signatures, group histories, and group encoding) and user-level features (user histories and user encoding). Table 1 below illustrates the results that we obtained in the ablation study.
>
> **Table 1: Impact of feature groups on prediction accuracy (Ablation Study).**
>
> | Feature Configuration | Features Used | Cov@25% | Cov@50% | Performance Gain |
> | :--- | :--- | :--- | :--- | :--- |
> | **1. Baseline** | Resources + Temporal | 14.9% | 28.8% | — |
> | **2. + Group** | Workload Signatures | 35.1% | 53.1% | **+20.2% (Primary Signal)** |
> | **3. + User** | User History (Full Model) | **45.8%** | **66.7%** | +10.7% (Secondary) |
>
> The baseline configuration using only resource and temporal features achieves 14.9% Cov@25%, establishing that job characteristics alone provide limited predictive power. Adding group-level features—including recurrence signatures, group execution histories, and workload encoding—yields a substantial gain of **+20.2 percentage points**, reaching 35.1% Cov@25%. This dramatic improvement confirms that workload recurrence and team behavior patterns constitute the **primary** predictive signal in datacenter job duration prediction. User-level features contribute an additional +10.6 percentage points (45.8% Cov@25%), demonstrating that individual user behavior provides meaningful but **secondary** value beyond group-level patterns.
>
> **3. Overfitting analysis**
>
> We conduct analyses to validate that our feature engineering generalizes beyond the training data and does not overfit to the test set, shown in Table 2 below.
>
> **Table 2: Generalization analysis on test set and unseen users.**
>
> | Analysis Type | Metric Comparison | Cov@25% Gap | Conclusion |
> | :--- | :--- | :--- | :--- |
> | **Overfitting** | Train (5-fold CV) vs. Test Set | 46.7% vs. 45.8% | **1.1%** | Negligible overfitting (Gap < 5%) |
> | **New Users** | Seen Users vs. **Unseen Users** | 45.9% vs. 40.1% | **5.8%** | Model generalizes to new users via Group features |
>
> First, we assess the train-test performance gap through 5-fold cross-validation on the training set. The minimal gap of approximately 1.0 percentage point indicates negligible overfitting. Second, we evaluate generalization to unseen users by partitioning the test set into jobs from users present in training (98.2% of test jobs) versus entirely new users (1.8%). Performance on seen users (45.9% Cov@25%) exceeds unseen users (40.1%) by 5.8 percentage points. While this gap reflects the expected advantage of user-specific features, the model maintains reasonable accuracy on unseen users, demonstrating robustness to distribution shift.

---

> ### Author Response · Authors · 2025-11-21
>
> **4. Cross-dataset generalization**
>
> For cross-dataset deployment with similar workloads but different users, our resource-based features and workload signatures transfer directly (35.1% Cov@25% as shown in the Ablation study), while user encoding require retraining on local data to achieve full performance (45.8%). Please note that the ablation study uses LightGBM only for simplicity. The prediction performance will be even better using our proposed powerful predictors.
>
>
> **Revisions to the Paper:**
>
> We have added a paragraph after the **Feature engineering** section to justify the ablation study findings:
>
> > *Ablation study. We use LightGBM to reveal that workload recurrence and group-level execution patterns are the dominant predictive feature over a resource-only baseline, while individual user behaviors provide secondary refinement. The results validate our benchmark design and demonstrate that all used features contribute meaningfully to prediction accuracy, with group-level patterns generalizable across users and resource features transferable across datasets.*
>
> We sincerely hope our answers address your questions. Thanks for your valuable comments on this work. We are grateful that the proposed prediction and scheduling benchmark can benefit the learning-augmented scheduling community.

---

> > ### Comment · Reviewer_ufpH · 2025-11-21
> > **Reviewer Response**
> >
> > Dear authors,
> >
> > Thank you for your responses to my questions and ablation study, this is a nice addition to the paper and addresses my main comment.  I will continue to support the paper.

---

### Official Review · Reviewer_aucU · 2025-10-30

**Soundness:** 3
**Presentation:** 3
**Contribution:** 2
**Rating:** 4
**Confidence:** 5

**Summary:**

This paper introduces Alibaba Trace for Learning-Augmented Scheduling (ATLAS), a research-ready dataset derived from Alibaba’s Platform of Artificial Intelligence (PAI) cluster trace—a production system that processes hundreds of thousands of ML jobs per day. The ATLAS dataset has been cleaned and features engineered to represent the inputs and constraints of non-clairvoyant scheduling, including user tags, resource requests (CPU/GPU/memory), and job structures with ground-truth processing times. This paper develops a prediction benchmark reporting prediction error metrics, along with feature importance analysis, and introduces a novel multiple-stage ML model.

**Strengths:**

The paper provides a real-world dataset and a standardized benchmark, which are valuable for studying learning-augmented scheduling in practical environments.

**Weaknesses:**

1.	Please clarify whether each job exclusively occupies its allocated resources or whether multiple jobs can coexist on the same resource group (e.g., node or GPU). This is important for understanding the dataset’s applicability to scheduling policies that handle interference and resource sharing.

2.	The trace seems to be collected under a specific scheduling policy. Please clarify what scheduler was used during data collection, and discuss whether this underlying policy could bias the dataset or affect the quality of downstream learned schedulers trained on this trace.

3.	The paper should elaborate on how preemptive scheduling is represented in the dataset. Are both preemptive and non-preemptive execution cases included? If so, how is preemption recorded and labeled?

**Questions:**

1.	Please clarify whether each job exclusively occupies its allocated resources or whether multiple jobs can coexist on the same resource group (e.g., node or GPU).

2.	Please clarify what scheduler was used during data collection, and discuss whether this underlying policy could bias the dataset or affect the quality of downstream learned schedulers trained on this trace.

3.	Are both preemptive and non-preemptive execution cases included? If so, how is preemption recorded and labeled?

---

> ### Author Response · Authors · 2025-11-21
>
> We thank the reviewer for their positive assessment of ATLAS as a valuable and research-ready benchmark. We particularly appreciate your insightful questions regarding resource sharing and underlying scheduler bias. These are critical for ensuring the dataset's correct applicability in real-world scenarios. Your feedback has encouraged us to significantly clarify the system model and data collection methodology in the revised paper. We address your specific questions below.
>
> **W1 and Q1. Resource sharing.**
>
> Thanks for this valuable feedback. We confirm that in the ATLAS dataset, multiple jobs can coexist on the same resource group, though this is rare. To clarify this, we recall that the underlying cluster uses the Fuxi scheduler with a **Reserving-and-Packing** policy for resource allocation:
>
> * **Exclusive Allocation (Reserving):** For high-GPU tasks (typically compute-intensive workloads such as BERT or ResNet training), the scheduler employs a reserving strategy. These tasks are prioritized and allocated exclusive access to high-end GPUs (e.g., V100s) to ensure performance isolation. Such jobs do not coexist on any resource group at any time.
> * **Shared Allocation (Packing):** For low-GPU tasks, which constitute the majority of the workload, the scheduler uses a packing strategy via fractional GPU sharing. This allows multiple task instances to share a single GPU by requesting fractional resources (e.g., a request in the range of 0 to 1).
>
> Trace analysis reveals that the median instance usage is only 0.042 GPUs (i.e., the majority are low-GPU tasks), making packing essential to prevent under-utilization. Furthermore, while resource sharing is possible, it is statistically infrequent: contention (heavy GPU utilization $\ge$ 95%) occurs in only 7% of the total time. Among these high-load scenarios, only 4.5% involve co-located instances (one compute unit running multiple instances simultaneously).
>
> In the **Job life-cycle** section (Line 197), we have following description. In production, PAI uses reserving-and-packing scheduling: it reserves high-end V100/V100M32 (NVLink) nodes for high-GPU or strict gang/locality tasks, and packs lower-GPU tasks onto T4/older ‘Misc’ machines via fractional-GPU sharing. We agree that an explicit discussion on resource sharing adds value to the paper. We have made the following changes in the revised version.
>
> **1. Adding statistical verfication**
> We added the following at the end of the **Job life-cycle** paragraph:
>
> > *Crucially, trace characterization verifies that GPU contention in this shared environment is statistically negligible: heavy GPU utilization ($\ge$ 95%) occurs in only 7% of cases, and among these high-load scenarios, only 4.5% involve co-located instances.*
>
> **2. Figure 2 Caption Update**
> We revised the caption of Figure 2 to explicitly mention fractional requests:
>
> > *The center panel shows planned versus actual resource usage, revealing users requesting more resources than they actually use. Instances can request fractional resources (e.g., 0.5 GPU), supporting resource sharing across multiple jobs.*
>
> We hope this addresses your question regarding resource sharing mechanics and frequency.
>
> **W2 and Q2. Scheduler bias in the dataset.**
>
> Thanks for this nice observation. We confirm that the dataset was collected under the Fuxi scheduler, which employs a reserving-and-packing policy [1]. We have noted this in the **Alibaba PAI–2020 Trace** paragraph (line 154) and **Job life-cycle** (line 195). We will explicitly clarify the implications of this policy on data bias and downstream learning.
>
> We identify three biases imposed by Fuxi's decisions and discuss their impact on learned schedulers:
>
> **(1) Allocation bias (Selection Bias).**
> Fuxi preferentially reserves high-end V100 nodes for compute-intensive tasks. Consequently, the dataset suffers from selection bias: it lacks counterfactual examples (e.g., heavy tasks running on low-end T4 GPUs). Without observing the performance difference of the same task across different GPUs, downstream models cannot learn the true hardware heterogeneity matrix (i.e., the speedup ratios). Instead, they merely overfit to Fuxi’s historical placement policy, mimicking the existing assignment logic rather than optimizing based on actual hardware capabilities.
>
> **(2) Gang-scheduling assembly bias.**
> Previous work [1] reports that 85% of instances require gang scheduling. The recorded job start times include the coordination latency required for Fuxi to allocate concurrent resources. Consequently, the job duration includes system-induced synchronization delays. A job taking 3 hours might include 30 minutes of unnecessary waiting for the last few instances to become available.

---

> ### Author Response · Authors · 2025-11-21
>
> **W2 and Q2. Scheduler bias in the dataset.**
>
> **(3) Interference bias.**
> This emerges from packing decisions: when Fuxi co-locates multiple instances on CPU machines, the resulting slowdowns from resource contention become part of the ground truth duration. Learning algorithms trained on this data will predict inflated durations. However, the primary constraint for placement is GPU type (V100 vs. T4) and quantity. CPU is considered a secondary resource, where contention arises as a derivative effect of this GPU-centric packing.
>
> **Impact on downstream quality and justification of trade-offs:**
>
> Defining the ground truth for job size presents a fundamental trade-off between **minimizing noise** and **ensuring scheduling robustness**.
>
> In the **Job processing time** paragraph (line 207), we noted that prior work [1] estimates job size by averaging instance durations. Taking the mean is less sensitive to gang-scheduling assembly bias. However, this method may also be too optimistic, as 1) **not all** instances follow gang scheduling and 2) it is unlikely that the longest-running instance completes at the expected run time. Such a measurement, therefore, almost always underestimates the true job's processing time.
>
> Our proposed fork-join method for defining ($p_j^{\star}$), on the other hand, is pessimistic, as it captures the full system reality. This is, however, not so harmful for production scheduling, because underestimated processing time, rather than overestimated, can cause propagated delays throughout the schedule. Thus, overestimation is slightly more favorable than underestimation [2, 3]. By maintaining the span from the first instance start to the last instance completion, we ensure that learned schedulers tend to overestimate job size and thus are robust for learning-augmented scheduling.
>
> To justify the bias in our job size estimation due to the underlying scheduling policy, we will add the following sentences after introducing two methods in **Job processing time** to justify the decisions we have made about our method for job size estimation.
>
> > *Admittedly, neither definition fully decouples the true job demand from Fuxi’s historical allocation decisions. However, the design trade-off prioritizes safety. Using the mean duration ($\bar{d}_t$) systematically underestimates occupancy, particularly for gang-scheduled tasks where the tail instance determines resource release. Conversely, our fork-join definition ($p_j^{\star}$) captures actual system constraints. We intentionally enforce this robustness to prevent learned schedulers from planning based on overly optimistic duration estimates.*
>
> **W3 and Q3. Preemptive vs. non-preemptive.**
>
> The ATLAS dataset records completed job executions without explicit preemption labels for both the scheduler level and the smallest execution unit (instance). During data collection, jobs in the production PAI cluster ran to completion once started (non-preemptive execution) [1]. Therefore, the trace dataset contains just the start and end timestamps for each instance.
>
> The preemptive scheduling algorithms mentioned in our LASched benchmark section (SRPT, PRR, etc.) are used only in our simulations. We replay the trace with different scheduling policies. The actual production trace reflects non-preemptive execution under Fuxi's scheduler. While researchers can simulate preemptive policies through trace replay as we did, the dataset itself does not contain preemption.
>
> We will mention this directly in the dataset description part **Alibaba PAI–2020 Trace**.
>
> > *Once started, jobs run to completion without preemption. The trace contains only start and end timestamps for each instance, and no suspension or resumption events.*
>
> We will also update Appendix, the table for small inference, distributed training, and large-scale jobs of ATLAS is demonstrated, showing that no preemption-related columns occur in the dataset.
>
> We sincerely appreciate your rigorous review and the time dedicated to evaluating our work. Your observations helped us identify these areas for clarification. We hope our responses and the updated manuscript fully address your concerns.
>
> **References**
>
> [1] Qizhen Weng, Wencong Xiao, Yinghao Yu, Wei Wang, Cheng Wang, Jian He, Yong Li, Liping Zhang, Wei Lin, and Yu Ding. "MLaaS in the Wild: Workload Analysis and Scheduling in Large-Scale Heterogeneous GPU Clusters." In *19th USENIX Symposium on Networked Systems Design and Implementation (NSDI 22)*, pp. 945-960. 2022.
>
> [2] Jun Woo Park, Alexey Tumanov, Angela Jiang, Michael A. Kozuch, and Gregory R. Ganger. "3Sigma: Distribution-based Cluster Scheduling for Runtime Uncertainty." In *Proceedings of the 13th EuroSys Conference (EuroSys '18)*, pp. 1-17. 2018.
>
> [3] Dan Tsafrir, Yoav Etsion, and Dror G. Feitelson. "Backfilling Using System-Generated Predictions Rather Than User Runtime Estimates." *IEEE Transactions on Parallel and Distributed Systems*, vol. 18, no. 6, pp. 789-803. 2007.

---

### Official Review · Reviewer_Lijb · 2025-10-30

**Soundness:** 3
**Presentation:** 3
**Contribution:** 3
**Rating:** 6
**Confidence:** 3

**Summary:**

The paper introduces ATLAS, a dataset derived from Alibaba's artificial intelligence platform cluster trace that processes many thousands of machine learning jobs per day. The dataset contains about 730K jobs with actual processing times in addition to many other metrics along with LASched, a benchmark for learning augmented scheduling. The work specifically addresses the gap between theoretical advances in learning augmented scheduling with the evaluation of those algorithms on real workloads. Specifically, they develop a procedure to reproducibly and comprehensively evaluate scheduling algorithms with clear train test splits. across three commonly used scheduling metrics including total completion time, max stretch and makespan.

**Strengths:**

- The scale of the dataset (730K) jobs and the vast metadata about the details of each job and resource it runs on make it a valuable contribution to the practical evaluation of learning augmented scheduling algorithms.
- Moreover, to the best of my knowledge, they seem to be the first to enforce leakage safe construction, that is only information that an actual scheduler would have, for the evaluation of their algorithms which is critically important for rigorous evaluation.
- Multiple scheduling objectives that are commonly used in the literature are evaluated on.
- Provide an easy to use simulator to evaluate various learning augmented scheduling algorithms.

**Weaknesses:**

- Excessive mathematical formalism and notation that is not used much after introduction. On page 4 and 5 for example, a fair bit of mathematical notation is introduced that would not look out of place in a scheduling paper with theoretical results. However, the notation is only used sparingly in pages 4, 5 and Algorithm 1 and there are no proofs in the paper that would benefit from such formal notation. If the notation was simplified the readability of the paper would significantly improve.
- The authors introduce a really powerful scheduling benchmark and evaluate on various algorithms but do not really discuss their performance in detail and why certain algorithms perform better than others.
- The authors do not really comprehensively evaluate the connection between prediction of the job duration and the actual scheduling algorithm used.

**Questions:**

- A major suggestion I have for the authors would be to simplify and/or reduce the amount of mathematical notation used here. While I understand that a major audience is the learning augmented scheduling researchers who tend to be more theoretical, in my personal opinion the notation here reduces accessibility to broader audiences (those implementing these systems in industry for instance) that this paper could impact.
- Another suggestion would be to really explain and elaborate why certain algorithms and job duration methods seem to perform better than others. There seems to be a disconnect in the writing between actually predicting the job duration versus the performance of the algorithms (competitive ratios). I could imagine that small changes in prediction could have large algorithmic impacts or even vice versa. Clearly characterizing this with a more in depth analysis connecting these two aspects would seriously improve the impact of the paper both for algorithm developers and those actually implementing schedulers in industry.
- Maybe I missed this but I don't think I see results showing that you used the various job duration prediction methods on all of the scheduling algorithms listed. Adding these results, even if it is only to the appendix would really help out both algorithm developers and those in industry.
- Given the scale and value of the dataset, I think it is reasonable for the authors to clearly explain what the holes are with current methods and suggest avenues/algorithmic directions for improvement. This seems low effort but would positively impact the community and improve uptake of this work.

If my concerns are addressed, I'm very inclined to increase my score. I think this paper adds substantial value to the community and if some of my questions are answered then it could have greater impact.

---

> ### Author Response · Authors · 2025-11-21
>
> We sincerely thank the reviewer for the very positive overall assessment and for highlighting the value of ATLAS and LASched for the learning-augmented scheduling community. We are especially grateful for the explicit statement that you would be inclined to increase your score if the concerns are addressed. Below, we respond to each weakness and question.
>
> **Q1. Improve accessibility and reduce math notations.**
>
> We appreciate the reviewer's feedback and agree that the notation density requires much explanation in the revised version. The original purpose of formalizing the PAI trace columns, as mentioned in Page 3, line 160, is to have unambiguous time semantics, label construction, and reproducibility checks.
>
> We will add, in the revised version, intuitive explanations alongside formalism to ensure accessibility for both theoretical researchers and industry researchers. In updated Appendix, individual notation and equations will be further explained with meanings and examples to ensure they are accessible to broader audiences.
>
> >For example, in Appendix the resource capacity constraint $\sum_i x_{i,m}(\tau) r_{t(i)} \leq c_m$ will be explained as the total resources used by all instances on a machine cannot exceed that machine's capacity. We make the following changes in the revised version.
>
> We also make following revised changes in the paper.
>
> **Job life-cycle**
>
> We add the following content about the demand vector with an example:
>
> > Each task t declares a per-instance demand vector $r_t = (r_{t,1}, r_{t,2}, r_{t,3})$ representing per-instance GPU, CPU, and memory requests. To be specific, a distributed training worker might request $r_t = (1, 8, 32)$ for 1 GPU, 8 CPUs, and 32GB memory.
>
> We add more details about the binary variable $x_{i,m}(\tau)$:
>
> > Let $x_{i,m}(\tau) \in \{0,1\}$ indicate that instance i is assigned to machine m at time $\tau$ (i.e., $x_{i,m}(\tau) = 1$ if instance i runs on m at time $\tau$, 0 otherwise).
>
> **Resource metrics and utilization**
>
> For this paragraph, we add one sentence at the beginning:
>
> > Let resource coordinates $k \in \{1,2,3\}$ denote GPU, CPU, and memory, respectively. We distinguish submit-time requests from post-execution utilization metrics.
>
> We revise several long mathematical formalizations to remove redundancy and improve the definition of utilization. For example, we change the sentence, $u_{cluster,k} = \frac{1}{H \cdot C_k}\sum_m \sum_{i} A_{i,k} = \frac{1}{H \cdot C_k}\sum_j A_{j,k}$, which makes resource–time conservation explicit: $\sum_m \sum_{i} A_{i,k} = \sum_t A_{t,k} = \sum_j A_{j,k}$, to:
>
> > $u_{cluster,k} = \frac{1}{H \cdot C_k}\sum_j A_{j,k}$.
>
> We make a total of 9 changes in this paragraph for better accessibility.
>
> Despite these changes, we respectfully maintain that mathematical precision might be necessary for three reasons.
>
> **(1) Bridging theoretical foundations:** Learning-augmented scheduling builds on extensive theory [1, 2, 3]. Our formalization enables theorists to verify whether algorithm assumptions (e.g., job arrival pattern) hold in ATLAS, making theoretical guarantees applicable to experimental results. Without this, the connection between theoretical papers and empirical validation remains ambiguous.
>
> **(2) Making 85-column data interpretable:** The raw Alibaba dataset has seven tables with a total of 85 columns. Our formalization clarifies job processing time reconstruction from timestamps using fork-join semantics: a job initiates parallel instances (fork) and holds resources until the slowest instance completes (join).  While prior studies might define job size simply by averaging instance durations, our formalization explicitly codifies the aggregation logic from the instance-task level. This disambiguation is essential for ensuring a reproducible benchmark.
>
> **(3) Enabling rigorous characterization:** Metrics like utilization $u_{cluster,k}$ establish a unified mathematical specification for workload analysis. Most studies reported aggregate statistics without derivation; our formalization eliminates ambiguity in how 'workload' is calculated. Future researchers can use it as a standardized rule to validate data integrity and objectively categorize workload regimes for algorithm selection.
>
> **References**
>
> [1] Manish Purohit, Zoya Svitkina, and Ravi Kumar. "Improving Online Algorithms via ML Predictions." In *Advances in Neural Information Processing Systems (NeurIPS)*, 2018.
>
> [2] Tianming Zhao, Wei Li, and Albert Y. Zomaya. "Learning-Augmented Scheduling." *IEEE Transactions on Computers*, 2024.
>
> [3] Silvio Lattanzi, Thomas Lavastida, Benjamin Moseley, and Sergei Vassilvitskii. "Online Scheduling via Learned Weights." In *Proceedings of the 2020 ACM-SIAM Symposium on Discrete Algorithms (SODA)*, pp. 1859-1877. 2020.

---

> ### Author Response · Authors · 2025-11-21
>
> **Q2. Compare each prediction method and algorithm and explain the performances.**
>
> We appreciate the feedback regarding the need for detailed justifications of prediction and scheduling performance.
>
> **1. Analysis of Prediction Methods (Page 10 Table 2)**
> Our results show distinct performance characteristics across the six methods:
>
> * **Meta-Stack & Two-Stage:** These achieve high rank correlation ($\rho$). Meta-Stack reduces variance through ensembling. Two-Stage handles skewed distributions via a classification-first approach, achieving the highest Cov@25 (%) by avoiding overfitting to medium-duration jobs.
> * **Isotonic & Recency:** These perform well but have limitations. Isotonic calibration improves confidence but cannot fix poor base predictions. Recency captures temporal drift but misses structural patterns.
> * **Recurring vs. All:** Recurring jobs (having historical instances) show consistently better prediction accuracy than the aggregate set.
> * **Overall Insight:** Calibration methods outperform Drift/History methods. This indicates that cross-sectional features (job types) matter more than temporal patterns in ATLAS, consistent with a shared cluster where diverse users run heterogeneous workloads.
>
> **Revised Text: Prediction Method Discussion**
> We have updated the discussion to the following comprehensive version:
>
> > *Table 2 details prediction performance. Two-Stage achieves the best coverage (60.8% Cov@25%, 80.4% Cov@50%) via its classification-first approach, despite similar rank correlations across Two-Stage, Meta-Stack, CQR, and Isotonic. The substantial gap between calibration-centric methods and the history-only HRAS baseline confirms the necessity of job-specific features. Recurring jobs, 81.3% of the test set, exhibit consistently higher accuracy, validating the utility of historical data. Although Meta-Stack offers marginally lower RMSLE, Two-Stage's superior coverage directly yields better scheduling performance.*
>
> **2. Relationship between Prediction and Scheduling**
> As noted in Section 4 (lines 472-475), high rank correlation explains why scheduling performs well despite imperfect duration predictions. We have refined the explanation to clarify how specific algorithms react to prediction errors:
>
> **Revised Text: Scheduling Sensitivity**
>
> > *High rank correlation ($\rho$) explains the outperformance of order-dependent algorithms despite prediction errors. SPJF achieves near-optimal total completion time because the objective prioritizes relative order over accurate prediction size; the 6.8% degradation from SJF to SPJF quantifies the specific cost of prediction error. Furthermore, SPJF outperforms PRR ($\lambda=0.7$), demonstrating that fully leveraging accurate predicted rankings supersedes partial usage. For makespan, LPPT shows moderate sensitivity, depending primarily on identifying the largest jobs. Preemption mitigates error impact by distributing delays across jobs or $m$ machines, enabling reasonable performance even with HRAS. In contrast, max-stretch is extremely sensitive to prediction quality: a single underestimated large job receives lower priority and accumulates queuing delays, severely degrading the worst-case metric.*
>
> We hope this response addresses your question about the comparison between prediction methods, scheduling algorithms and prediction quality impacts.
>
> **Q3. Use various prediction methods on scheduling algorithms.**
>
> Thanks for suggesting including the results for all prediction methods applied to all scheduling algorithms.
>
> In the revised version, we have significantly expanded the original paper's Table 3 to provide a comprehensive benchmark across the three primary scheduling objectives, resulting in 36 additional scheduling algorithm-predictor combinations. The obtained results share shown in Table shown below (next comment). Conclusions from updated table:
>
>
> **(1) Two-Stage and Meta-Stack consistently outperform baselines**
> Across all three objectives, the proposed Two-Stage and Meta-Stack models consistently yield the best scheduling metrics (highlighted in bold).
>
> **(2) High rank correlation drives scheduling performance**
> Prediction accuracy alone is insufficient; ordering is critical for size-based policies like SRPT and SPJF. Two-Stage and Meta-Stack achieve high Spearman rank correlations, ensuring short jobs are rarely misprioritized. Conversely, HRAS's weaker correlation ($\rho=0.772$) results in significantly degraded performance.
>
> **(3) Objective-specific algorithm dominance**
> The benchmark results align with classical scheduling theory, validating the data. For the makespan objective, predictors paired with LPPT consistently outperform those paired with SPPT. This is theoretically expected, as LPT is a known approximation for makespan minimization on parallel machines, whereas SPT performs poorly. Capturing this behavior demonstrates the benchmark's reliability for evaluating non-clairvoyant algorithms.

---

> ### Author Response · Authors · 2025-11-21
>
> Results table for **Question 3** is shown below.
>
> **Table 1: Comprehensive performance evaluation of scheduling algorithm-predictor combinations across three objectives (lower is better).**
>
> | (A) Max-Stretch Algo | $\rho_{S,max}$ | $\rho_{S,99}$ | $\rho_{S,med}$ | (B) T.C.T. Algo | Ratio | (C) Makespan Algo | Ratio |
> | :--- | :--- | :--- | :--- | :--- | :--- | :--- | :--- |
> | **OPT (EDF at $S^*$)** | **1.000** | **1.000** | **1.000** | **SRPT** | **1.000** | **LPT** | **1.000** |
> | SRPT | 1.189 | 1.150 | 0.900 | SJF | 1.001 | SPT | 1.539 |
> | LAS/FB | 1005.85 | 607.79 | 155.62 | RR | 1.975 | Greedy | 1.452 |
> | | | | | FIFO | 5.372 | Random | 1.955 |
> | CQR-SPRPT | 15.69 | 3.88 | 0.121 | CQR-SPJF | 1.075 | CQR-LPPT | 1.517 |
> | CQR-EDF(pred) | 4287.21 | 1214.92 | 14.65 | CQR-PRR | 1.252 | CQR-SPPT | 1.694 |
> | HRAS-SPRPT | 39.80 | 24.59 | 0.235 | HRAS-SPJF | 1.823 | HRAS-LPPT | 1.874 |
> | HRAS-EDF(pred) | 5036.06 | 1474.90 | 20.01 | HRAS-PRR | 1.929 | HRAS-SPPT | 1.801 |
> | Iso-SPRPT | 15.58 | 3.62 | 0.127 | Iso-SPJF | 1.087 | Iso-LPPT | 1.500 |
> | Iso-EDF(pred) | 4437.21 | 1233.32 | 14.99 | Iso-PRR | 1.265 | Iso-SPPT | 1.615 |
> | Meta-SPRPT | **14.97** | **3.35** | 0.125 | Meta-SPJF | 1.072 | Meta-LPPT | 1.529 |
> | Meta-EDF(pred) | 4342.12 | 1232.13 | 14.83 | Meta-PRR | 1.252 | Meta-SPPT | 1.692 |
> | TwoSt-SPRPT | 15.63 | 3.64 | **0.119** | TwoSt-SPJF | **1.066** | TwoSt-LPPT | **1.498** |
> | TwoSt-EDF(pred) | 4346.84 | 1212.56 | 14.68 | TwoSt-PRR | 1.246 | TwoSt-SPPT | 1.604 |
> | Rec-SPRPT | 17.64 | 3.78 | 0.120 | Rec-SPJF | 1.097 | Rec-LPPT | 1.568 |
> | Rec-EDF(pred) | 4567.88 | 1292.03 | 16.34 | Rec-PRR | 1.278 | Rec-SPPT | 1.769 |
>
> **Q4. Future directions.**
>
> **1. Tail sensitivity and outliers.**
> Max-Stretch results reveal that current methods fail at the worst-case tail behavior required for fairness; a single poor prediction destroys the metric. Future research must move from average-case risk minimization to Distributionally Robust Optimization (DRO) or CVaR. Specifically, researchers should investigate Hard Example Mining to dynamically re-weight losses towards high-stretch jobs.
>
> **2. Feature representation mismatch.**
> Complex models (HRAS) sometimes underperform simple ones (Two-Stage) by overfitting absolute durations rather than relative ordering. Future work on rank-driven algorithms should focus on learning representations that capture relative rank.
>
> **3. Prediction-scheduling objective mismatch.**
> Minimizing Mean Squared Error (MSE) suits total completion time (e.g., Two-Stage+SPJF) but fails for max-stretch. Researchers should explore asymmetric loss functions or differentiable scheduling layers.
>
> **Revised Conclusion:**
>
> > LAShed serves as a comprehensive benchmark for non-clairvoyant scheduling, revealing that while standard predictors are sufficient for aggregate metrics like total completion time, they fail to address tail-sensitive objectives. By exposing these gaps, our dataset identifies three concrete directions for future algorithmic development: (1) asymmetric loss functions to address prediction-scheduling objective mismatches, (2) distributionally robust optimization (e.g., CVaR) to minimize worst-case stretch, and (3) rank-aware feature learning to prioritize relative ordering over absolute accuracy.
>
> We sincerely appreciate your detailed reviews. We hope our responses address your concerns and merit a re-evaluation of our submission.

---

> > ### Comment · Reviewer_Lijb · 2025-11-25
> > **Response to Authors**
> >
> > Thank you to the authors for their hard work. The paper is a lot more accessible now and I'm happy to increase my score.

---

### Author Response · Authors · 2025-11-26
**Update file and comprehensive response to all reviewers**

Dear Area Chair and Reviewers,

We sincerely thank you for your constructive feedback. We have uploaded a revised manuscript with all major changes highlighted in dark purple.

**Summary of Key Revisions:**

1. Enhanced clarity and accessibility: We have streamlined the introduction to improve accessibility and explicitly clarified for the broad ML community. We have also expanded Appendix A with comprehensive notation tables and formal equations. We also clarified about job characteristics, such as resource sharing and non-preemptive nature of jobs, in Section 2.

2. Rigorous validation: To address concerns regarding model robustness, we have added a detailed ablation study and overfitting analysis in Section 3.1. These experiments confirm that our model generalizes well and does not overfit to the training data.

3. Comprehensive benchmarking: We have significantly expanded the scheduling results (Table 4). Rather than relying on a default predictor, we now evaluate the downstream scheduling performance of all proposed prediction methods

**Response to Specific Concerns:**

We have posted detailed, point-by-point responses to each reviewer’s individual comments below. We hope that these revisions fully address the questions raised. We are willing to answer any further questions.

We invite you to review the revised paper and our responses.

Sincerely,

The Authors

---

### Author Response · Authors · 2025-11-30
**Global Response to Area Chair**

Dear Area Chair,

We understand and appreciate that a new Area Chair has been assigned to our submission. To assist your assessment, we summarize our contributions, the strong reviewer consensus, and the specific revisions made to address all concerns.

---

**1. Our contributions**

The paper addresses a fundamental barrier in learning augmented scheduling: the absence of public, research-ready datasets with ground-truth processing times for rigorous empirical evaluation. We provide ATLAS, the first research-ready dataset derived from Alibaba's PAI cluster traces.

(1) Scale and Reality: Over 730,000 jobs with complete resource profiles and verified ground-truth processing times $p^*_j$, features absent from existing production traces.

(2) Prediction and Scheduling Benchmark: LASched is the first standardized benchmark that uses "submit-time only" feature engineering, preventing the information leakage violating non-clairvoyant scheduling assumptions.

(3) Methodological Advance: Our novel Two-Stage prediction model outperforms traditional ML prediction methods by ~20% in coverage (60.8% Cov@25%) by leveraging job recurrence patterns.


**2. Reviewer Consensus**

Reviewers strongly endorsed these contributions, agreeing that a standardized benchmark for this field is overdue and that ATLAS successfully fills a critical gap between theoretical assumptions and system realities. They praised the shift away from synthetic data, noting that the dataset's scale and ground-truth labels serve as a valuable bridge to practical applications. Furthermore, the leakage-safe construction was highlighted as a methodological necessity; reviewers confirmed that our strict exclusion of post-execution metrics establishes the first benchmark to ensure true non-clairvoyant conditions. Reviewers also commended LASched's complete ecosystem (simulator, reproducible baselines) as a plug-and-play resource enabling direct algorithm comparison.


**3. Resolution of Technical Concerns**

All revisions are highlighted in **dark purple** in updated manuscript.

**A. Formalization and Accessibility (Section 2.1, Appendix A, Introduction)**
- *Concern:* One reviewer noted that mathematical formalization is without sufficient justification; the other observed introduction assumes familiarity with learning-augmented algorithms.
- *Resolution:* We made explicit justifications for formalization: (i) bridging theoretical foundations so theorists can verify algorithm assumptions, (ii) disambiguating 85-column raw data with precise semantics, and (iii) enabling rigorous workload characterization. We explain notations in Sec 2.1 and in Appendix A with tables and easy-to-understand examples, and we revised the whole introduction for accessibility to the broader ML community.


**B. System Realism: Resource Sharing and Scheduler Bias (Section 2.1)**
- *Concern:* Questions regarding resource sharing, preemption, and scheduler bias.
- *Resolution:* We detailed the Reserving-and-Packing policy: high-GPU tasks receive exclusive allocation while low-GPU tasks use fractional sharing, with co-location occurring in only 4.5% of high-load scenarios. We documented three bias sources (allocation, gang-scheduling assembly, interference) and justified our fork-join definition as intentionally conservative—overestimation is preferable to underestimation for robust scheduling. We clarified that jobs run to completion without preemption. Preemptive algorithms in LASched are evaluated through trace-replay simulation only.

**C. Feature Robustness: Ablation and Overfitting Analysis (Section 3.1)**
- *Concern:* Whether User/Group features lead to overfitting and limit generalizability.
- *Resolution:* We added ablation studies showing group-level recurrence is the dominant (+20.2% gain over resource-only baseline), while user features provide secondary refinement. Overfitting is negligible (1.1% gap between cross-validation and test). For unseen users, the model maintains reasonable accuracy; for cross-dataset deployment, resource and group features transfer directly without retraining.

**D. Expanded Scheduling Benchmark (Section 4, Table 4)**
- *Concern:* The connection between prediction error and scheduling outcomes needed elaboration.
- *Resolution:* We expanded Table 4 to 36 predictor-scheduling combinations across three objectives. Key insights: (i) rank correlation ($\rho$=0.95), not absolute accuracy, drives performance for order-dependent algorithms; (ii) Two-Stage consistently outperform baselines across all objectives; (iii) max-stretch is extremely sensitive to prediction quality, revealing limitations of current methods for tail-sensitive objectives.


**4. Conclusion**

ATLAS and LASched provide the foundation required to advance learning-augmented scheduling from theory to practice. We believe this work contributes significantly to the community. Also, authors have addressed all concerns, referring to following discussions.

Sincerely,

Authors

---

### Meta-Review · Area_Chair_ZGJm · 2026-01-07

**Summary:**

This paper proposes a critical benchmark and the work is well executed. The reviewers' concerns have mostly been addressed after author rebuttal.

Before rebuttal, the reviewers have concerns around problem formulation, simplified benchmark setting, and model overfitting. The authors have:

* improved writing and added intuitive examples
* provided evidence to justify the system model
* added additional ablation studies around feature robustness
* expanded Table 4 to include 36 predictor-scheduling combinations

**Reviewer Concerns:**

The outstanding concerns are the limited time span of the benchmark and lack of considering of more complex environments.

**Reviewer Scores:**

Reviewer aucU would likely to increase the score.

---

### Decision · Program_Chairs · 2026-01-26

Accept (Poster)